# Improved base editing and functional screening in *Leishmania* via co-expression of the AsCas12a ultra variant, a T7 RNA polymerase, and a cytosine base editor

**Nicole Herrmann May[1], Anh Cao[1], Annika Schmid[1], Fabian Link[1,2], Jorge Arias-del-Angel[1,2], Elisabeth Meiser[1], Tom Beneke[1]\***

[1]Department of Cell and Developmental Biology, Biocentre, University of Würzburg, Am Hubland, Würzburg, Germany; [2]Division of Immunology, Paul-Ehrlich-Institut, Langen, Germany

## eLife Assessment

This **important** article describes a meticulously-developed improved strategy for generation of functionally null mutants in Leishmania spp. via cytosine base editing, with reduced background toxicity and enhanced efficiency relative to a previously-described method. The authors show use of the strategy in a small-scale loss-of-function screen, providing **compelling** evidence that large-scale screens will be possible. The newly developed tools will be of great interest to researchers working with Leishmania and beyond.

**\*For correspondence:**
tom.beneke@uni-wuerzburg.de

**Competing interest:** The authors declare that no competing interests exist.

**Abstract** The ability to analyze the function of all genes in a genome is highly desirable, yet challenging in *Leishmania* due to a repetitive genome, limited DNA repair mechanisms, and lack of RNA interference in most species. While our introduction of a cytosine base editor (CBE) demonstrated potential to overcome these limitations (Engstler and Beneke, 2023), challenges remained, including low transfection efficiency, variable editing rates across species, parasite growth effects, and competition between deleterious and non-deleterious mutations. Here, we present an optimized approach addressing these issues. We identified a T7 RNAP promoter variant ensuring high editing rates across *Leishmania* species without compromising growth. A revised CBE single-guide RNAs (sgRNAs) scoring system was developed to prioritize STOP codon generation. Additionally, a triple-expression construct was created for stable integration of CBE sgRNA expression cassettes into a *Leishmania* safe harbor locus using AsCas12a ultra-mediated DNA double-strand breaks, increasing transfection efficiency by ~400-fold to 1 transfectant per 70 transfected cells. Using this improved system for a small-scale proof-of-principle pooled screen, we successfully confirmed the essential and fitness-associated functions of CK1.2, CRK2, CRK3, AUK1/AIRK, TOR1, IFT88, IFT139, IFT140, and RAB5A in *Leishmania mexicana*, demonstrating a significant improvement over our previous method. Lastly, we show the utility of co-expressing AsCas12a ultra, T7 RNAP, and CBE for hybrid CRISPR gene replacement and base editing within the same cell line. Overall, these improvements will broaden the range of possible gene editing applications in *Leishmania* species and will enable a variety of loss-of-function screens in the near future.

## Introduction

CRISPR/Cas9 gene editing has greatly improved loss-of-function experiments in *Leishmania*, and bar-seq screens, which involve individually deleting, barcoding, and pooling mutants for analysis, have facilitated the functional dissection of large gene cohorts (*Baker et al., 2021*; *Beneke et al., 2019*; *Beneke et al., 2019*; *Beneke and Gluenz, 2020*; *Beneke et al., 2017*; *Burge et al., 2020*; *Damianou et al., 2020*). However, the applications of bar-seq screens are limited, and it remains difficult to target the vast *Leishmania* repertoire of repetitive genetic elements by gene replacement approaches. In addition, *Leishmania* parasites lack crucial components of the non-homologous DNA end joining (NHEJ) pathway (*Passos-Silva et al., 2010*; *Zhang et al., 2022*). As a result, CRISPR-induced DNA double-strand breaks (DSBs) lead to unpredictable DNA deletions, increased cell death during DNA repair failures, prolonged repair times, and generally low editing rates (*Zhang et al., 2017*; *Zhang and Matlashewski, 2015*; *Zhang and Matlashewski, 2019*; *Zhang et al., 2022*). Without other means of selection, such as the selection of drug-resistance-associated edits, this complicates further the targeting of dispersed multi-copy genes and applications of pooled CRISPR transfection formats. While RNA interference (RNAi) could offer an alternative (*de Paiva et al., 2015*; *Lye et al., 2022*), its use is confined to species within the *Viannia* subgenus, restricting its applicability in the majority of *Leishmania* species (*Lye et al., 2010*; *Ullu et al., 2004*).

We have recently shown how the use of the hyBE4max cytosine base editor (CBE) (*Zhang et al., 2020*) could potentially overcome these limitations. We demonstrated how the conversion of cytosine to thymine, and thereby the introduction of STOP codons, enables the functional disruption of single- and multi-copy genes in *Leishmania* species without requiring DSB, homologous recombination, the additional use of donor DNA, or isolation of clones (*Engstler and Beneke, 2023*). Importantly, we presented how this method can be used to identify essential genes in pooled loss-of-function screens via the delivery of plasmid libraries, thereby providing a proof-of-principle experiment for how large-scale pooled CRISPR transfection fitness screens could be carried out in *Leishmania*. However, we also highlighted the need for additional improvements of our method (*Engstler and Beneke, 2023*).

For example, since we used a ribosomal promoter derived from *Leishmania donovani* for expression of the CBE and single-guide RNA (sgRNA), editing efficiencies varied greatly between species, and for some *Leishmania* parasites, sufficient editing could be only achieved after weeks in culture. Our attempts to use different promoters, such as the T7 RNA polymerase (RNAP) promoter, were unsuccessful, as their employment for CBE sgRNA expression resulted in strong growth defects. Furthermore, the transfection of pooled plasmid libraries led to combinatorial knockout effects, with approximately one-third of all transfectants harboring more than one CBE-sgRNA expression plasmid. In addition, the transfection efficiency of plasmid pools was generally low, yielding only ~1 transfectant out of 30,000 transfected cells. To compensate for combinatorial effects and the relatively low transfection rate, a large number of cells would need to be transfected to enable large-scale pooled CRISPR transfection screens. Lastly, we noticed that non-deleterious mutations within the CBE editing window can become dominant over desired mutations, especially when targeting essential or growth-affecting genes.

Here, we aimed to develop our base editing approach further and deliver needed improvements. We show how we have identified a T7 RNAP promoter variant that enables stable expression of CBE sgRNAs and results in high editing rates within a short period of time without having significant effects on parasite growth. In addition, we present a construct that allows to express not only T7 RNAP and the hyBE4max CBE but also a highly efficient variant of a Cas12a nuclease derived from *Acidaminococcus* sp. (AsCas12a ultra [*Zhang et al., 2021*], formerly AsCpf1). Using AsCas12a ultra-mediated DSBs, we have developed a novel approach to deliver and integrate CBE sgRNA expression cassettes into the *Leishmania* 18S rRNA safe harbor locus (*Misslitz et al., 2000*; *Sörensen et al., 2003*). We present evidence that the integration of these expression cassettes is precise at the target locus and we demonstrate that this approach has increased transfection rates by up to 400-fold. Using this improved CRISPR tool, we conducted a small-scale loss-of-function screen in *L. mexicana*, targeting nine known essential genes with 24 CBE sgRNAs and 15 non-targeting control sgRNAs. This approach successfully detected all growth-associated phenotypes in a pooled screening format. In a final optimization step, we have also improved our CBE sgRNA design and scoring tool, https://www.leishbaseedit.net/, to prioritize sgRNAs with fewer theoretical risks of generating edits that do not result in a STOP codon. We believe that these optimization steps now enable us to deliver a range of gene

editing applications, including large-scale loss-of-function screens, in *Leishmania* species without limitations due to extreme cases of gene copy numbers, aneuploidy, and/or lack of RNAi components.

## Results and discussion

### T7 RNAP promoter variant-driven CBE sgRNA expression reduces toxicity and improves editing efficiency

In our first version of a CBE toolbox for *Leishmania*, we utilized a ribosomal promoter derived from *L. donovani* for CBE protein and sgRNA expression. While achieving high editing rates with this single plasmid system (*Figure 1A*; *Engstler and Beneke, 2023*), efficiencies varied greatly among species, requiring for example over 40 days of incubation to reach significant editing in *Leishmania major* parasites. We previously attempted to minimize these species-specific variations by segregating the CBE expression from the CBE sgRNA cassette in a dual-construct system and by employing a T7 RNAP promoter for CBE sgRNA expression (*Figure 1B*). However, upon transfection of CBE sgRNA expression constructs into *L. major* T7 RNAP- and CBE-expressing parasites, the doubling time increased from 7 to about 14 hr, highlighting a significant growth defect through T7 RNAP-driven CBE sgRNA expression. On the contrary, we observed higher editing rates in a shorter period of time compared to our approach with the single plasmid ribosomal promoter system, indicating potential advantages (*Engstler and Beneke, 2023*).

Here, our first aim was to eliminate this growth defect observed in the dual-construct system while maintaining high editing rates. Assuming that the growth defect resulted from excessive CBE sgRNA expression, we wondered whether modifying the T7 RNAP promoter sequence and altering the transcription initiation site could alleviate the observed toxicity. We therefore introduced a variety of CBE sgRNA expression plasmids into our previously established *L. major* cell line, expressing CBE and T7 RNAP from the β-tubulin locus, as well as tdTomato from the 18S rRNA locus (*Engstler and Beneke, 2023*; *Figure 1B*). To drive CBE sgRNA expression, we thereby utilized two T7 RNAP promoter variants (T7 T-10 and T7 G-10), known for their ability to reduce transcription activity in the closely related parasite *Trypanosoma brucei* (*Wirtz et al., 1998*). We also manipulated the transcription initiation site by altering the number of guanines (either two or one), a factor believed to impact transcription yield (*Padmanabhan et al., 2020*; *Figure 1C*). Additionally, we compared CBE sgRNA expression via T7 RNAP promoters with the ribosomal promoter derived from *L. donovani*, either in a single- or dual-construct system setup. For our measurements, we transfected CBE sgRNA expression plasmids with two tdTomato-targeting CBE sgRNAs. While the 'target' CBE sgRNA was designed to yield an early STOP codon through cytosine-to-thymine (C-to-T) conversion, the 'control' CBE sgRNA was designed to induce a C-to-T conversion that results in a neutral substitution (codon change but no amino acid change). Following transfection, we then analyzed growth rates (*Figure 1D*) and tdTomato reporter signals in the resulting non-clonal populations (*Figure 1E and F*). Our aim was to identify the promoter variant with the highest activity and least toxicity.

As anticipated, when utilizing the ribosomal promoter for CBE sgRNA expression, the dual-construct system resulted in a more significant reduction of the tdTomato reporter signal compared to the single plasmid system. The complete knockdown of the tdTomato reporter was also achieved using all T7 RNAP promoter variants with the dual-construct system. While there were no detectable differences in these knockdown efficiencies (*Figure 1E and F*), there were notable variations in the growth rates (*Figure 1D*). Consistent with our prior observations (*Engstler and Beneke, 2023*), cells transfected with constructs containing the unmodified T7 RNAP promoter sequence (T7wt GG in *Figure 1D*) exhibited an almost twofold increase in doubling time. On the contrary, employing for CBE sgRNA expression the T7 T10 GG, T7 G10 GG, and ribosomal promoter resulted in no significant growth rate increase. The doubling times here ranged from 7.5 to 8.1 hr. While this aligned with measurements obtained for cells transfected with our single plasmid system (*Figure 1D*), the complete knockdown of the tdTomato reporter was much more rapid using any dual-construct system (*Figure 1E*). Specifically, the complete depletion of the tdTomato reporter signal was achieved in just 7 days (*Figure 1E*) compared to 33 days when using the same CBE sgRNA in *L. major* parasites previously (*Engstler and Beneke, 2023*). Compared to our first toolkit version (*Engstler and Beneke, 2023*), this demonstrates a major improvement in our method to generate loss-of-function mutants in non-clonal populations.

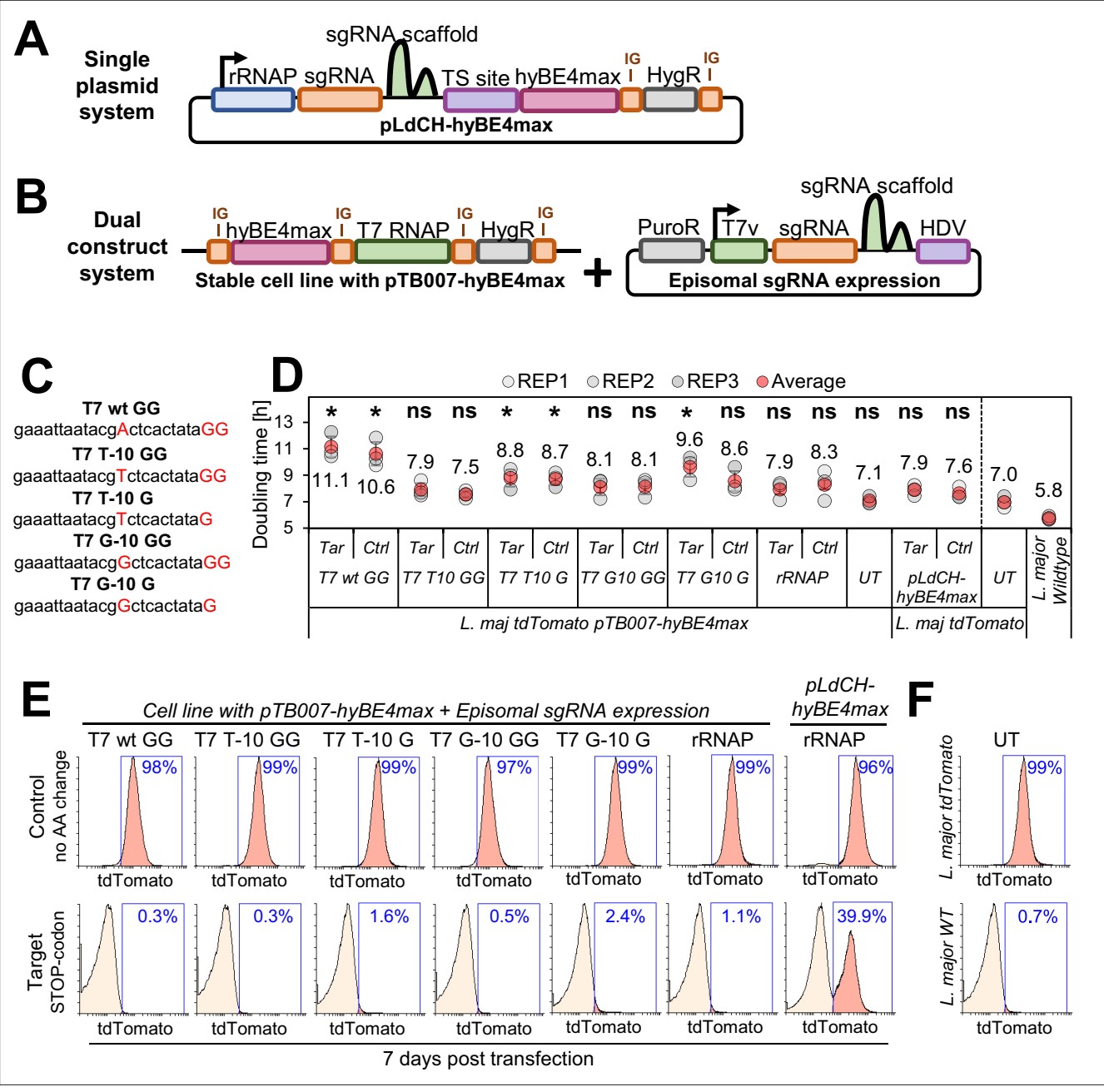

**Figure 1.** Optimization of cytosine base editor (CBE) single-guide RNA (sgRNA) expression in *L. major*. Schematics of base editing strategies to transfect *Leishmania* parasites either with a (**A**) single or (**B**) dual base editing system. (**A**) Plasmid pLdCH-hyBE4max (**Engstler and Beneke, 2023**) contains (from left to right) a *L. donovani*-derived ribosomal RNA (rRNA) promoter, sgRNA expression cassette, hepatitis delta virus (HDV) ribozyme containing transsplice sequence (TSS), hyBE4max CBE, *L. donovani*-derived A2 intergenic sequence, and hygromycin-resistance marker. (**B**) Plasmid pTB007-hyBE4max (**Engstler and Beneke, 2023**) contains separated through intergenic regions (from left to right) a hyBE4max CBE, T7 RNA polymerase (RNAP), and hygromycin-resistance marker. This construct is integrated into the β-tubulin locus of *L. major* parasites (LmjF.33:339,096-341,104). The CBE sgRNA expression vector contains a puromycin-resistance marker, T7 RNAP promoter, sgRNA expression cassette, and HDV ribozyme. (**C**) Different versions of the T7 RNAP promoter for CBE sgRNA expression have been tested. (**D**) Doubling times of tdTomato-expressing *L. major* transfected non-clonal populations with the single plasmid system and variants of the dual plasmid system for targeting of tdTomato. Error bars show standard deviations of triplicates. Tar: Cells transfected with a CBE sgRNA expression plasmid that facilitates the introduction of a STOP codon within the tdTomato open-reading frame (ORF); Ctrl: Cells transfected with a CBE sgRNA expression plasmid that causes a codon mutation without amino acid change. Asterisks indicate Student's t-test: *p>0.05. (**E**) FACS plot of parasites shown in (**D**), analyzed 7 days post transfection. (**F**) FACS plot showing tdTomato-expressing and wildtype *L. major* parasites.

## DSBs introduced by AsCas12a ultra increase integration rates of donor DNA constructs

Having successfully improved editing efficiencies, while at the same time reducing the toxicity of CBE sgRNA expression and targeting, we next attempted to increase transfection rates of the CBE sgRNA expression constructs. Previously, we have shown that ~30,000 *L. mexicana* wildtype cells need to be transfected with our single plasmid system (*Figure 1A*) to obtain one transfectant (*Engstler and Beneke, 2023*). Since genomic dropout screens need to be performed at a high representation of about 500 cells per sgRNA construct (*Sanjana et al., 2014*; *Yau and Rana, 2018*), this relatively low transfection rate would make any large-scale screen challenging. In addition, we wanted to enable screening of base editor libraries over long periods and in complex environments, such as in vivo experiments, without risking that parasites would lose their CBE sgRNA plasmid, vary their plasmid copy number or transfer plasmids between cells. Therefore, we wanted to integrate CBE sgRNA expression constructs into a safe harbor locus. Strategies to increase transfection rates and stably integrate expression cassettes by homologous recombination have been previously developed in *T. brucei*. Here, the use of an I-SceI mega- (*Glover and Horn, 2009*) or zinc finger- (*Schumann et al., 2017*) nuclease-induced DSB at a safe harbor locus has been shown to increase the transfection efficiency for the integration of RNAi expression constructs significantly. In combination with improved transfection protocols (*Burkard et al., 2007*; *Glover and Horn, 2009*), this has enabled RNAi loss-of-function screens on a genome-wide scale (*Glover et al., 2015*; *Horn, 2022*; *Schumann Burkard et al., 2011*). Alternatively, it has been shown in *Leishmania* that CRISPR/Cas9-induced DSBs can increase the rate of correct donor DNA integration, regardless of homology flank length (*Beneke et al., 2017*).

To develop a strategy for stable integration of CBE sgRNA expression constructs in *Leishmania* species, we decided to employ a Cas12a nuclease variant from *Acidaminococcus* species. Specifically, we generated an expression construct (pTB107 or pTB106) to co-express in *Leishmania* the AsCas12a ultra variant (*Zhang et al., 2021*), a T7 RNAP, and the hyBE4max CBE (*Engstler and Beneke, 2023*; *Figure 2A*). The use of AsCas12a ultra had several advantages compared to other approaches mentioned above. First, compared to the use of I-SceI meganuclease (*Glover and Horn, 2009*), it allowed us to select any TTTV-PAM containing locus of interest to insert our CBE sgRNA expression construct, simply by just changing the Cas12a crRNA sequence. Second, Cas12a editing – just like any CRISPR strategy – is much simpler to design than a zinc finger nuclease (*Schumann et al., 2017*) or TALENs approach. Third, compared to a functional SpCas9 nuclease, there was no risk of interference between the Cas9 base editor and Cas9 nuclease sgRNAs. Finally, since AsCas12a has an intrinsic RNase activity that allows processing of its own crRNA array (*Paul and Montoya, 2020*), multiplexing of guide sequences is possible, which could expand the scope of standard CRISPR knockin and knockout approaches in *Leishmania* in future studies. For example, Cas12a crRNA arrays with four or more guides can be assembled and transfected to introduce multiple DSBs within one gene. Since Cas12a generates sticky DNA ends that facilitate recombination via microhomology-mediated end joining and homologous recombination (*Zhang et al., 2021*), this approach could effectively disrupt target genes without requiring the addition of donor DNA and this may provide an alternative approach to our here presented base editing method in the future. Moreover, CBE sgRNAs could be multiplexed by interspacing them with Cas12a direct repeats (DRs), enabling simultaneous targeting of multiple genes in one cell. Ultimately, we figured that the co-expression of the AsCas12a ultra nuclease, a T7 RNAP, and the CBE would generate a hybrid gene editing toolbox that has many possible applications.

To introduce AsCas12a-mediated DSBs, we decided to adapt the LeishGEdit approach, which was previously developed for in vivo transcription of Cas9 sgRNAs (*Beneke and Gluenz, 2019*; *Beneke et al., 2017*). Specifically, we designed two overlapping oligos capable of being mutually amplified, generating a Cas12a crRNA template DNA (*Figure 2B*). This resulting template featured an unmodified T7 RNAP promoter sequence, facilitating the in vivo transcription of the Cas12a crRNA. To ensure high binding affinity of AsCas12a with its Cas12a crRNA, we used an optimized DR from *DeWeirdt et al., 2021*. Subsequently, this enabled AsCas12a ultra-mediated DSBs for donor DNA integration. To test this approach, we utilized a donor DNA construct containing mNeonGreen (mNG) as a reporter and blasticidin as a resistance marker, allowing for a relatively straightforward assessment of integration accuracy and transfection rate (*Figure 2A*). Additionally, we varied the length of homology

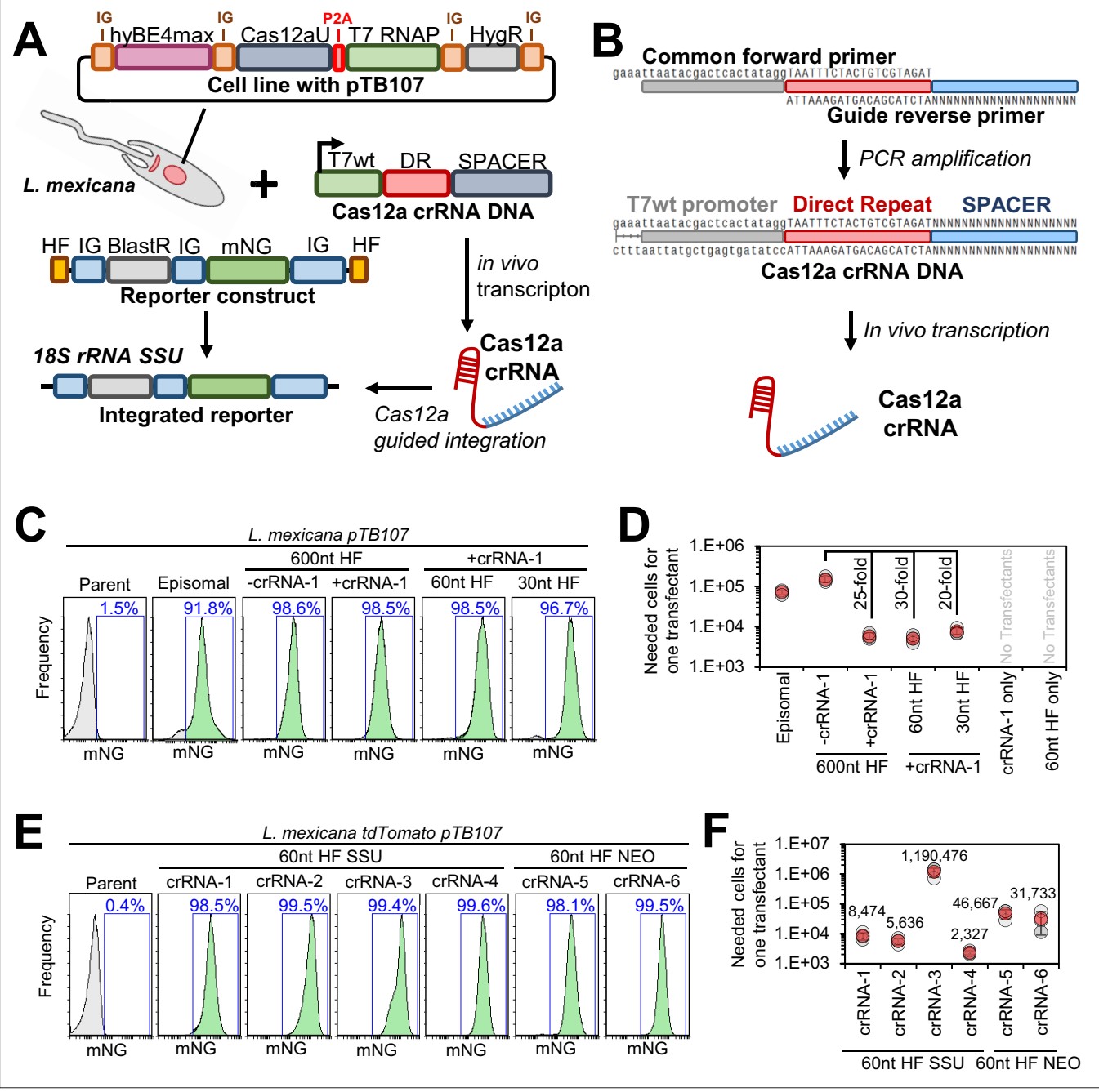

**Figure 2.** Developing an AsCas12a-mediated integration system. (**A**) Schematic of the Cas12a-mediated integration approach. A stable *L. mexicana* cell line maintains plasmid pTB107 as an episome and is co-transfected with a donor DNA reporter construct and a Cas12a crRNA template DNA. Using a T7 RNA polymerase (RNAP) promoter, the Cas12a crRNA template DNA is in vivo transcribed to facilitate a Cas12a-mediated double-strand break (DSB) at the 18S rRNA SSU locus. Using homology flanks the reporter construct is efficiently integrated at the site of the DSB. Plasmid pTB107 thereby allows for the expression of hyBE4max cytosine base editor (CBE), AsCas12a ultra (*Zhang et al., 2021*), T7 RNAP, and hygromycin-resistance marker. AsCas12a ultra and T7 RNAP are expressed as a fusion protein but then cleaved through a P2A self-cleaving peptide. The reporter constructs consist of a pPLOT plasmid tagging cassette (*Beneke et al., 2017*) with additional homology flanks, allowing for the expression of mNeonGreen and a blasticidin-resistance marker. (**B**) Schematic of the Cas12a crRNA template DNA generation. Two overlapping primers are amplified in a PCR. The forward primer is common and contains an unmodified T7 RNAP promoter and an optimized Cas12a direct repeat variant (DR) (*DeWeirdt et al., 2021*). The guide reverse primer contains the crRNA-specific spacer sequence aligning to the targeted locus and the Cas12a DR to enable the overlap with the forward primer. (**C and E**) FACS plot showing *L. mexicana* pTB107 non-clonal populations following the transfection described in (**A**). Percentages represent the remaining proportion of mNeonGreen-expressing cells. (**C**) The homology flank (HF) length used for reporter construct integration has been varied and the additional use of Cas12a crRNA was tested. The transfection of a circular plasmid, containing the reporter construct, was included as a control

*Figure 2 continued on next page*

*Figure 2 continued*

(episomal). (**E**) Six different Cas12a crRNAs have been tested using two different integration loci (as described in the main text). (**D and F**) Efficiency of transfections shown in (**C and E**) were measured. The number of cells required for the transfection to obtain one transfectant is shown. Error bars show standard deviations of triplicates.

The online version of this article includes the following source data and figure supplement(s) for figure 2:

**Figure supplement 1.** Verification of AsCas12a-mediated reporter construct integration.

**Figure supplement 1—source data 1.** Raw DNA images of *Figure 2—figure supplement 1B and E* with labels.

**Figure supplement 1—source data 2.** Raw DNA images of *Figure 2—figure supplement 1B and E* without labels.

flanks to investigate whether Cas12a-mediated integration of donor DNA remained independent of homology flank length, as previously observed with Cas9 (*Beneke et al., 2017*). Using homology flanks routinely used for integration into the 18S rRNA safe harbor locus of various *Leishmania* species (*Misslitz et al., 2000*; *Sörensen et al., 2003*), we then designed one Cas12a crRNA targeting the 200 bp region between both homology regions (*Figure 2—figure supplement 1A*). To evaluate if Cas12a could enhance the transfection rate, we compared transfections of these reporter constructs with and without the addition of Cas12a crRNA.

Like findings with Cas9, we observed that Cas12a-induced DSBs in *L. mexicana* increased the rate of correct donor DNA integration, irrespective of homology flank length. Integration of reporter constructs with 600 nt homology flanks showed no difference in reporter signal (*Figure 2C*), but the addition of Cas12a crRNA boosted transfection efficiency by approximately 25-fold when assessing blasticidin-resistance rates (*Figure 2D*). When reducing homology flank length to 60 nt, we observed no impact on the reporter signal or transfection rate when Cas12a crRNA was present during transfections. However, in the absence of Cas12a crRNA, no transfectants were obtained, confirming that homologous recombination of the reporter is only independent of homology flank length if a Cas12a-mediated DSB has occurred. Further reducing the homology flank length to 30 nt did not significantly decrease transfection efficiency. However, the percentage of mNG positive cells decreased from ~99% in all other samples to 96.7% (*Figure 2C*), likely due to imprecise recombination events in some cells. Conversely, Sanger sequencing the integration sites of these non-clonal populations indicated flawless recombination for all constructs (*Figure 2—figure supplement 1A, B, and C*). Interestingly, diversity in the reporter signal was also noted when transfecting the mNG construct as an episome (*Figure 2C*), possibly pointing toward plasmid copy number variants between transfected cells. In addition, episomal transfections using the same number of DNA molecules resulted in a significantly lower number of transfectants (*Figure 2D*), concluding the advantages of integrating expression constructs via Cas12a-mediated DSBs.

We then wondered whether employing different Cas12a crRNAs could enhance the transfection efficiency even further. Three additional Cas12a crRNAs within the 18S rRNA SSU locus were selected for testing (*Figure 2—figure supplement 1D*). In addition, we sought to examine the integration into a 'landing-pad', such as a previously introduced resistance marker. We expected that this would further increase the accuracy and efficiency of the transfection. We therefore decided to test our system in a tdTomato-expressing *L. mexicana* cell line (*Engstler and Beneke, 2023*) and designed two Cas12a crRNAs to integrate mNG-blasticidin donor DNA constructs within their neomycin-resistance marker (*Figure 2—figure supplement 1D*). Given that 60 nt homology flank length was sufficient for accurate and efficient integration of donor DNA (*Figure 2C and D*), we decided to test all six Cas12a crRNAs in combination with donor DNA constructs containing 60 nt homology flanks. This meant that homology flanks could be easily adapted through long-primer PCR without the need for bacterial cloning.

Across all six Cas12a crRNAs tested, no significant changes in the proportion of mNG-expressing cells were observed when analyzing non-clonal populations post transfection (*Figure 2E*). However, there was a substantial variation in the rate of blasticidin-resistant cells, indicative of Cas12a crRNA efficiency (*Figure 2F*). For Cas12a crRNA-1, we confirmed our previous results, yielding 1 transfectant per ~8500 transfected cells. However, other Cas12a crRNAs, particularly Cas12a crRNA-4, performed significantly better, resulting in 1 transfectant out of approximately 2300 transfected cells. Interestingly, despite being only 44 bp upstream of Cas12a crRNA-4, Cas12a crRNA-3 exhibited a 500-fold lower integration efficiency. Surprisingly, both neomycin-targeting Cas12a crRNAs were less efficient

than the majority of 18S rRNA SSU locus-targeting guides (*Figure 2F*). Given the exceptional efficiency of Cas12a crRNA-4, we verified the integration of donor DNA at its site in the non-clonal population through Sanger sequencing, confirming flawless integration (*Figure 2—figure supplement 1D, E, and F*).

## Integration of CBE sgRNA expression cassettes via AsCas12a ultra-introduced DSBs increases editing rates

After successfully validating our here developed AsCas12a ultra-mediated integration system, we then decided to combine our Cas12a knockin system with our optimized CBE sgRNA expression cassette (*Figure 3A*). For this purpose, we developed two CBE sgRNA expression plasmids, namely pTB104 and pTB105, each incorporating a puromycin-resistance marker and the most effective T7 RNAP promoter variant (T7 T10 GG) for CBE sgRNA expression (as identified in assays above; *Figure 1C, D, and E*). While pTB104 featured 350 nt homology flanks adjacent to the neomycin-targeting Cas12a crRNA-6, pTB105 included 350 nt homology flanks adjacent to our best-performing Cas12a crRNA-4, targeting the 18S rRNA SSU locus. This design allowed for the integration of both CBE sgRNA expression cassettes with and without the addition of Cas12a crRNAs. To assess if such a system could effectively generate a functional null mutant through C-to-T conversion, we chose to target tdTomato again and tested a tdTomato-targeting CBE sgRNA in *L. donovani*, *L. mexicana*, and *L. major*. Parasites expressed tdTomato, along with the CBE, Cas12a ultra, and T7 RNAP (*Figure 3A*). Additionally, we investigated whether the efficiency of the tdTomato knockdown would be influenced by how the CBE sgRNA expression cassette was integrated. We therefore compared Cas12a crRNA-4 and 6 in combination with 60 and 350 nt homology flanks.

Strikingly, we observed a complete depletion of the tdTomato signal just 7 days post transfection in all tested species when integrating CBE sgRNA expression cassettes into the 18S rRNA SSU locus. This occurred irrespective of homology flank length. However, integrating cassettes into the neomycin-resistance marker ('landing-pad approach') resulted in only partial tdTomato knockdowns in *L. major* and *L. donovani*, while a complete knockdown could again be achieved in *L. mexicana*. Although we have not further investigated the cause for this, we assume that either the low efficiency of Cas12a crRNA-6 or the number of tdTomato-neomycin-expression cassettes caused this discrepancy. Overall, this highlights that integrating CBE sgRNA expression cassettes into the 18S rRNA SSU locus using Cas12a crRNA-4 in combination with plasmid pTB105 would yield the highest CBE sgRNA editing efficiency. We therefore tested next the transfection rate of CBE sgRNA expression cassettes using Cas12a crRNA-4 for integration in *L. mexicana*, *L. major*, and *L. donovani*. We reached high transfection efficiencies, yielding up to one transfectant out of ~470 transfected *Leishmania* parasites, with *L. donovani* being the most and *L. major* being the least efficient (*Figure 3C*, left panel).

We then wondered whether we could increase the transfection efficiency even further by using the Lonza Nucleofector technology. Previous studies in *T. brucei* demonstrated that the improved transfection rate achieved through I-SceI meganuclease-mediated cleavage could be further increased when combined with the Lonza Nucleofector technology, resulting in up to 1 transfectant per 100 transfected cells (*Glover and Horn, 2009*). Considering this, we explored whether this approach would be as effective in *Leishmania* parasites. We therefore conducted a side-by-side comparison of the transfection efficiency between the Lonza Nucleofector Basic Parasite kit and the *Trypanosoma*-optimized buffer (Tb-BSF buffer, see Materials and methods section for further details) (*Schumann Burkard et al., 2011*), both in combination with an Amaxa Nucleofector 2b electroporator. To increase the number of parasites per transfection even further, we compared our standard transfection protocol using $5 \times 10^6$ cells per transfection with up-scaled transfection formats involving $1 \times 10^8$ cells and $2.5 \times 10^8$ cells per electroporation. Testing these conditions in *L. mexicana*, we confirmed that the Lonza Nucleofector technology could indeed boost transfection efficiency by an additional 20-fold (*Figure 3C*, right panel). When transfecting $1 \times 10^8$ cells per transfection with 10 µg of PacI-linearized pTB105 plasmid DNA, we achieved up to one transfectant out of 70 transfected cells (*Figure 3C*, right panel). This represents a more than 400-fold increase in transfection efficiency compared to our previous rates (*Engstler and Beneke, 2023*). Moreover, it demonstrates that a CBE sgRNA library with 40,000 constructs could be transfected at a representation rate of 500-fold using just $20 \times 10^8$ cells across 20 transfection cuvettes. Ensuring a 500-fold library representation represents a substantial improvement compared to previous RNAi screens in *T. brucei*, where screens typically achieved about five- to

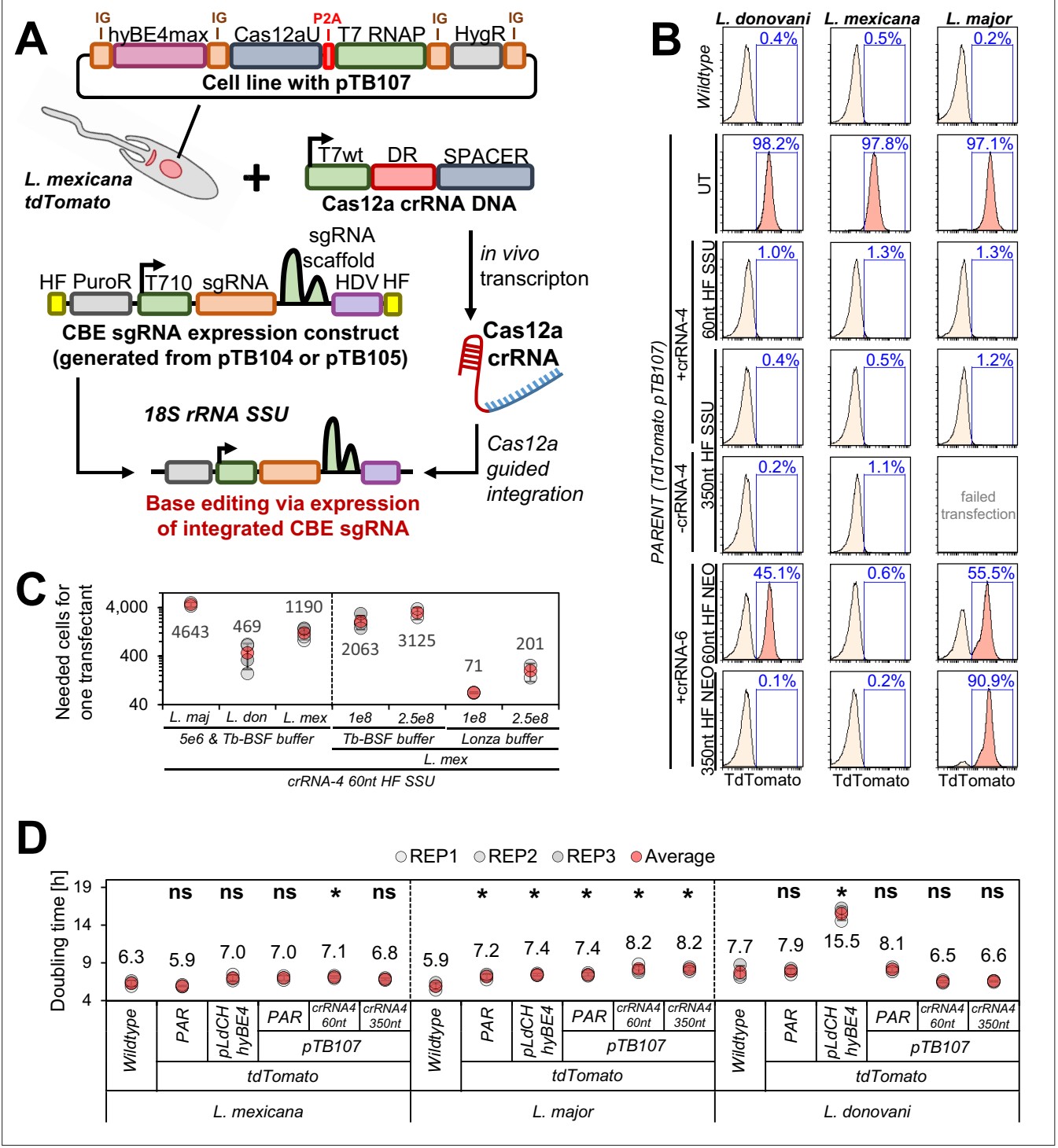

**Figure 3.** Integration of cytosine base editor (CBE) single-guide RNA (sgRNA) expression cassettes via AsCas12a ultra. (**A**) Schematic of Cas12a-mediated integration of the CBE sgRNA expression construct. The pTB107 stable cell line is co-transfected with two constructs: (1) a Cas12a crRNA template DNA and (2) a CBE sgRNA expression construct. The Cas12a crRNA template DNA is in vivo transcribed and consists of an unmodified T7 RNA polymerase (RNAP) promoter (T7wt, light green), a Cas12a direct repeat (DR, red) and a 20 nt Cas12a guide target sequence (SPACER, blue). The CBE sgRNA expression construct is integrated into the 18S rRNA SSU locus, following the Cas12a-mediated double-strand break (DSB). This donor construct contains two homology flanks (HF, yellow), a puromycin-resistance marker (gray), a T7 T-10 GG promoter (green), a guide target sequence (orange), a Cas9 scaffold (dark green), and an HDV (purple). (**B**) FACS plots show pTB107 parasites that express tdTomato and have been transfected with pTB104 and pTB105 CBE sgRNA expression cassettes, containing a tdTomato-targeting guide. Percentages represent the remaining proportion of non-clonal tdTomato-expressing cells. (**C**) Following dilutions after transfection, the number of puromycin-resistant transfectants obtained per

*Figure 3 continued on next page*

*Figure 3 continued*

transfected cell was calculated. Error bars show standard deviations of triplicates. (**D**) Doubling times for transfected *Leishmania* parasites shown in (**B**). PAR: parental cell line; UT: Un-transfected control.

ninefold library representation (*Glover et al., 2015*; *Horn, 2022*; *Morris et al., 2002*; *Schumann Burkard et al., 2011*). This high representation rate is considered crucial for large-scale dropout screens, as it can affect hit identification by minimizing variations between replicates. However, it's worth noting that drug-resistance screens can also be conducted at much lower coverage (*Sanjana et al., 2014*; *Yau and Rana, 2018*).

While the significantly enhanced transfection rates are a central aspect of our improved base editing method presented here, it is crucial to emphasize that these improvements were achieved without impacting parasite growth. Through an assessment of doubling times in transfected and parental cell lines, we confirmed that the expression of CBE, T7 RNAP, AsCas12a ultra, and the CBE sgRNA had minimal or no impact on parasite growth across all three species (*Figure 3D*). This marks a substantial improvement, considering that the expression of CBE and/or CBE sgRNA in our first toolbox version, using the pLdCH-hyBEmax plasmid, could lead to significant growth defects in some *Leishmania* species (*Engstler and Beneke, 2023*).

Having established this optimized system, we next decided to target PF16, a gene encoding a central pair protein of the axoneme that has been demonstrated to be essential for *Leishmania* flagellar motility (*Beneke et al., 2017*; *Engstler and Beneke, 2023*; *Martel et al., 2017*). For our test, we chose to utilize a previously employed PF16-targeting CBE sgRNA, namely PF16-3. This specific CBE sgRNA induces paralysis in *Leishmania* parasites by introducing a thymidine homo-polymer ('TTTTT') within the PF16 coding sequence (CDS). While our usual design for CBE sgRNAs aims to introduce STOP codons through C-to-T conversion, we selected PF16-3 for this test because its editing activities are known to vary across species (*Engstler and Beneke, 2023*). In earlier attempts, we had to express CBE sgRNA PF16-3 in non-clonal populations for 42 days to achieve sufficient editing in *L. major* when using our pLdCH-hyBE4max single vector (*Figure 1A*; *Engstler and Beneke, 2023*). Meanwhile, in *L. donovani*, high editing rates were already achieved 14 days post transfection. Now, we expected that editing rates had improved uniformly across all species.

As anticipated, we observed a significant increase in editing activity in *L. major* parasites. Just 14 days post transfection, we found that the homo-polymer had been fully introduced in non-clonal populations when using our T7 RNAP promoter-based dual vector system, while no editing could be observed using the ribosomal promoter single-vector system at this time point (*Figure 4B*, *L. major* panel). This was also reflected in the analysis of mutant swimming speed, revealing clear motility defects in respective transfectants (*Figure 4A*, *L. major* panel).

We next proceeded to test our optimized system also in *L. mexicana* and *L. donovani* parasites, once again employing Cas12a for integration of the CBE sgRNA expression construct. However, here we chose to target PF16 in two cell lines: (i) one that possessed the pTB107 construct and a tdTomato reporter, and (ii) one that possessed the pTB107 construct only (WT pTB107). Additionally, we took this opportunity to compare motility and CBE editing rates in cells that had been transfected with Cas12a crRNAs that target the neomycin-resistance marker (Cas12a crRNA-6) and the 18S rRNA SSU locus (Cas12a crRNA-4).

Our results clearly showed that just 6 days post transfection, the PF16 target site was fully edited, and the overwhelming majority of all cells were paralyzed in all tested scenarios (*Figure 4A and B*). However, surprisingly editing slowly reversed in *L. mexicana* cells harboring the pTB107 construct only (*Figure 4A and B*). Although we decided not to further investigate the reason for this unexpected result, we assumed that it might be caused by a small proportion of cells where the CBE sgRNA expression cassette was incorrectly integrated. This could potentially result in the silencing of the CBE sgRNA expression or even lead to the deletion of the guide cassette. Since the PF16 mutation in *L. mexicana* is known to be associated with a mild growth defect (*Beneke et al., 2019*), PF16-deficient mutants would eventually be outcompeted over time, leading slowly to the reversal of editing rates. While this escape or reversal of mutant phenotypes might initially appear problematic for detecting growth-associated or other phenotypes in pooled screens, it is important to note that this phenomenon has also been observed with other loss-of-function tools. For instance, RNAi escape has been reported in *T. brucei* when targeting growth-associated genes (*Ariyanayagam et al., 2005*; *Schlecker*

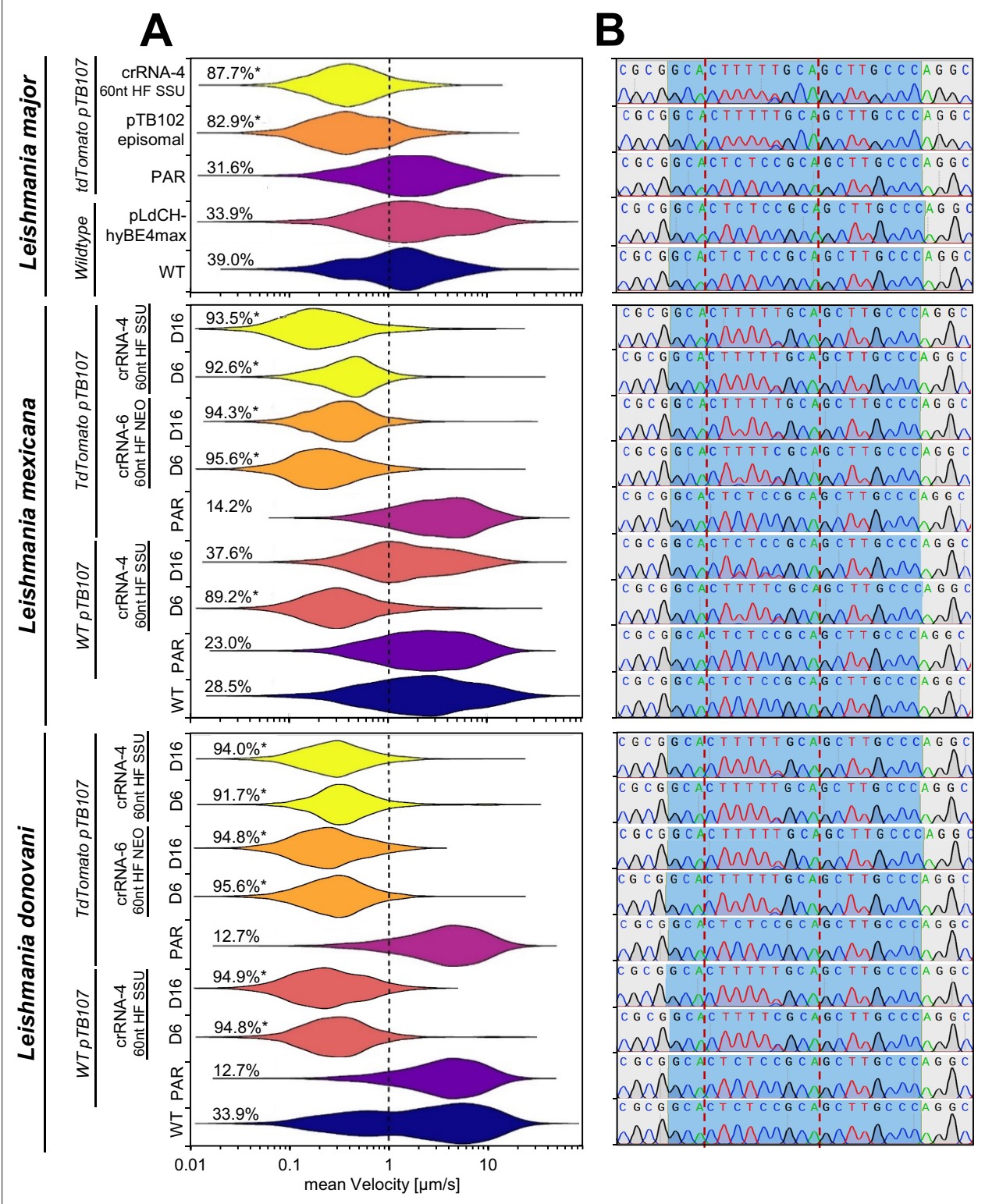

**Figure 4.** Targeting PF16 via Cas12a-delivered cytosine base editor (CBE) single-guide RNA (sgRNA) expression cassettes. *L. major*, *L. mexicana, L. donovani* wildtype, and tdTomato/pTB107 cell lines were transfected with a range of CBE sgRNA expression constructs in order to functionally mutate PF16. Different strategies for construct delivery and sgRNA expression were tested, including the integration of constructs into the neomycin-resistance marker (using Cas12a sgRNA-6) and the 18S rRNA SSU locus (Cas12a sgRNA-4), each using 60 nt homology flanks. For comparison to our previous system (**Engstler and Beneke, 2023**), we also expressed CBE sgRNAs from an episome and transfected pLdCH-hyBE4max into *L. major*. (**A**) Violin plot of pooled replicates from motility tracked non-clonal populations. The mean velocity of tracked cells was plotted 14 days post transfection for *L.*

*Figure 4 continued on next page*

*Figure 4 continued*

*major,* and 6 and 16 days post transfection for *L. mexicana* and *L. donovani*. The total percentage of tracked cells showing a velocity of less than 1 µm/s is highlighted. Each population was analyzed using a Cramér-von Mises test to detect any shift in the population distribution toward lower speed. Percentages are marked with an asterisk when that shift was significant (*p>0.05). (**B**) Corresponding Sanger sequencing trace plots. Blue shading: 20 nt guide target sequence. Red dotted lines: hyBE4max editing window.

The online version of this article includes the following figure supplement(s) for figure 4:

**Figure supplement 1.** Oxford Nanopore Technology (ONT) and Illumina sequencing of non-clonal transfectants.

**Figure supplement 2.** Blast analysis of Oxford Nanopore Technology (ONT) sequencing.

**Figure supplement 3.** Sequencing of clones isolated from an AsCas12a-mediated library transfection.

**Figure supplement 4.** Novel scoring for cytosine base editor (CBE) single-guide RNA (sgRNA) sorting.

*et al., 2005*). Nevertheless, genome-wide loss-of-function screens have been successfully performed in these parasites (*Alsford et al., 2011*). But more importantly the reversal of the PF16 mutation was not observed in any other tested cell line when comparing editing and motility rates 6 and 16 days post transfection. Both tested *L. donovani* cell lines and the tested *L. mexicana* cell line that possessed the pTB107 construct and the tdTomato expression cassette (*Figure 4A and B*) remained paralyzed over the entire observation time. To confirm that these cells would have no potential ever to reverse their phenotype, we decided to Illumina sequence these non-clonal paralyzed *L. mexicana* PF16 mutant populations 16 days post transfection. Analyzing their sequencing data, we did not find a single read of an unedited PF16 CDS (with approximately 35× genome coverage), confirming that indeed all cells within this transfected non-clonal population were completely mutated (*Figure 4—figure supplement 1A*).

## Cas12a-mediated DSB ensures the integration of one CBE sgRNA per *L. mexicana* transfectant

We next wanted to verify how exactly the CBE sgRNA expression cassette had been integrated via Cas12a-mediated DSB in these non-clonal *L. mexicana* populations that possessed the pTB107 construct and the tdTomato reporter, as well as the PF16-targeting CBE sgRNA expression construct 16 days post transfection. Since the tdTomato expression cassette is just ~400 nt upstream of the Cas12a crRNA-4 target sequence, we wondered whether it had come to any obvious interference between the tdTomato and CBE sgRNA expression cassettes. Given the highly repetitive nature of the 18S rRNA SSU locus, we employed Oxford Nanopore Technology (ONT) sequencing to determine the exact integration pattern of both cassettes. For the analysis of sequencing reads, we mapped them against two customized genomes. In the first genome, we assumed that the tdTomato expression cassette was integrated separately from the CBE sgRNA construct, implying that each cassette had integrated on the opposite allele (*Figure 4—figure supplement 1B*). In the second genome, we assumed that both cassettes were integrated on the same allele and could be detected on a single ONT read (*Figure 4—figure supplement 1C*).

Interestingly, we could only map unique ONT reads to the first scenario, where cassettes were assumed to be integrated on separate alleles and present in only one copy in the genome. These ONT reads covered the entire tdTomato or CBE sgRNA expression cassette, as well as sequences adjacent to homology flanks used for integration (*Figure 4—figure supplement 1B and C*).

We also used our ONT data to validate whether the tdTomato and CBE sgRNA expression cassettes (*Figure 4—figure supplement 2A*) had integrated into any unintended genetic loci. The CDS of the tdTomato reporter gene and the sgRNA expression cassette, comprising the optimized T7 RNAP promoter (T7 T10 GG), PF16-3 targeting spacer, and Cas9 sgRNA scaffold sequence, were analyzed using a standard nucleotide BLAST search (*Figure 4—figure supplement 2B*). Then, all matched ONT reads were extracted and we verified whether the full expression cassettes (*Figure 4—figure supplement 2A*) were correctly integrated at the intended position within the 18S rRNA SSU locus. As expected, all identified cassettes in the extracted contigs were correctly integrated (*Figure 4—figure supplement 2C and D*). Importantly, no ONT reads containing the tdTomato or CBE sgRNA cassettes could be linked to any other locus. Moreover, no contigs contained multiple or both, the tdTomato and CBE sgRNA expression cassettes, suggesting again that these cassettes had integrated into opposite alleles and are present in only one copy in the genome.

To validate our findings, we then revisited our Illumina sequencing data and analyzed the coverage of relevant genetic features. Specifically, we normalized all reads to the total number of reads per chromosome and compared the normalized coverage of reads mapped to the CDS of PF16, the CBE sgRNA cassette, the CDS of tdTomato, and the CDS of the neomycin-resistance marker. While we found equal read coverage of PF16-mapped reads and the chromosome harboring PF16, read coverage for the CBE sgRNA cassette, the CDS of tdTomato, and the CDS of the neomycin-resistance marker were approximately half of the read coverage of their harboring chromosome 27 (*Figure 4— figure supplement 1D*). This further confirmed our hypothesis that indeed the CBE sgRNA expression cassette had only been integrated on one allele, and most likely, this meant that only one CBE sgRNA cassette copy was present per *L. mexicana* cell. Given that the recognition site of Cas12a crRNA-4 is contained within the homology flank used for tdTomato integration at the 18S rRNA locus, this may contribute to the integration pattern we observe. But since the homology sequences are identical to the original sequences at the locus, the reasons to why this affects the integration of the CBE sgRNA expression cassettes remain unclear.

Nevertheless this represents an improvement since, in our previously used single-vector episomal system, approximately one-third of all *L. mexicana* transfectants possessed more than one CBE-sgRNA expression plasmid, potentially causing combinatorial knockout effects (*Engstler and Beneke, 2023*). To verify that in our new dual system each cell indeed harbored only one CBE sgRNA, we decided to test if isolated clones from a transfected library would possess one or multiple CBE sgRNA sequences. In addition to testing this in *L. mexicana*, we also wanted to assess this for *L. donovani* and *L. major*, as we had not sequenced respective mutants via ONT and Illumina sequencing.

Using again Cas12a crRNA-4 for the integration of a small-scale pTB105 CBE sgRNA expression cassette library (containing 13 different CBE sgRNA sequences), we isolated 10 clones from each species, PCR amplified the integrated CBE sgRNA locus, and sequenced it by Sanger sequencing. As expected, we found only unique sequencing reads for all tested *L. mexicana* clones, confirming the integration of only one guide per transfectant (*Figure 4—figure supplement 3*). However, for *L. donovani* and *L. major*, 2 out of 10 sequencing reads showed mixed guide sequences (*Figure 4— figure supplement 3*), suggesting that multiple CBE sgRNAs can be integrated per cell. Furthermore, the integration of multiple sgRNAs per cell remained random. While testing other Cas12a crRNAs in the future may identify a more unique integration locus for *L. donovani* and *L. major* parasites, any compensatory effects through the integration of multiple CBE sgRNAs in these species will likely be minimal and presumably not detectable in large-scale screens with a high library representation rate. Alternatively, a 'landing-pad' for integrating CBE sgRNA expression cassettes could be used, such as the neomycin-resistance gene targeted by the Cas12a crRNA-6 presented here (*Figure 2F*, *Figure 2—figure supplement 1D*, and *Figure 4*). Therefore, we believe that this method is suitable not only for screens in *L. mexicana* but also in *L. donovani* and *L. major*.

## Detection of fitness-associated phenotypes in a pooled loss-of-function screen

To provide evidence that our improved CRISPR base editing system can indeed detect a range of mutant phenotypes in pooled screens, including fitness-associated phenotypes, and to confirm significant improvements over our previous method (*Engstler and Beneke, 2023*), we next performed a small-scale loss-of-function fitness screen in *L. mexicana* parasites. First, we pooled 39 CBE sgRNAs and cloned them into pTB105, the vector designed for integration into the 18S rRNA locus using the best-performing Cas12a guide, crRNA-4. Of these sgRNAs, 24 were designed to introduce a STOP codon within the first half of open-reading frames (ORFs) essential for survival or normal proliferation in promastigote culture. Targeted genes included five essential kinases for which null mutants could not be generated in individual studies or in the kinome-wide bar-seq screen carried out by *Baker et al., 2021*. These included casein kinase 1 isoform 2 (CK1.2) (*Rachidi et al., 2014*), cdc2-related kinase 2 and 3 (CRK2, CRK3) (*Duncan et al., 2016*; *Hassan et al., 2001*; *Yagoubat et al., 2020*), Aurora kinase 1 (AUK1/AIRK) (*Chhajer et al., 2016*), and target of rapamycin kinase (TOR) 1 (*Madeira da Silva and Beverley, 2010*). We also included sgRNAs targeting the Ras-related protein (Rab) 5a, believed to be essential as null mutants could not be generated (*Rastogi et al., 2016*), and sgRNAs targeting three intraflagellar transport (IFT) proteins (IFT139, IFT140, and IFT88), all predicted to result in growth-associated phenotypes (*Beneke et al., 2019*; *Beneke et al., 2024*; *Sunter et al., 2018*). As

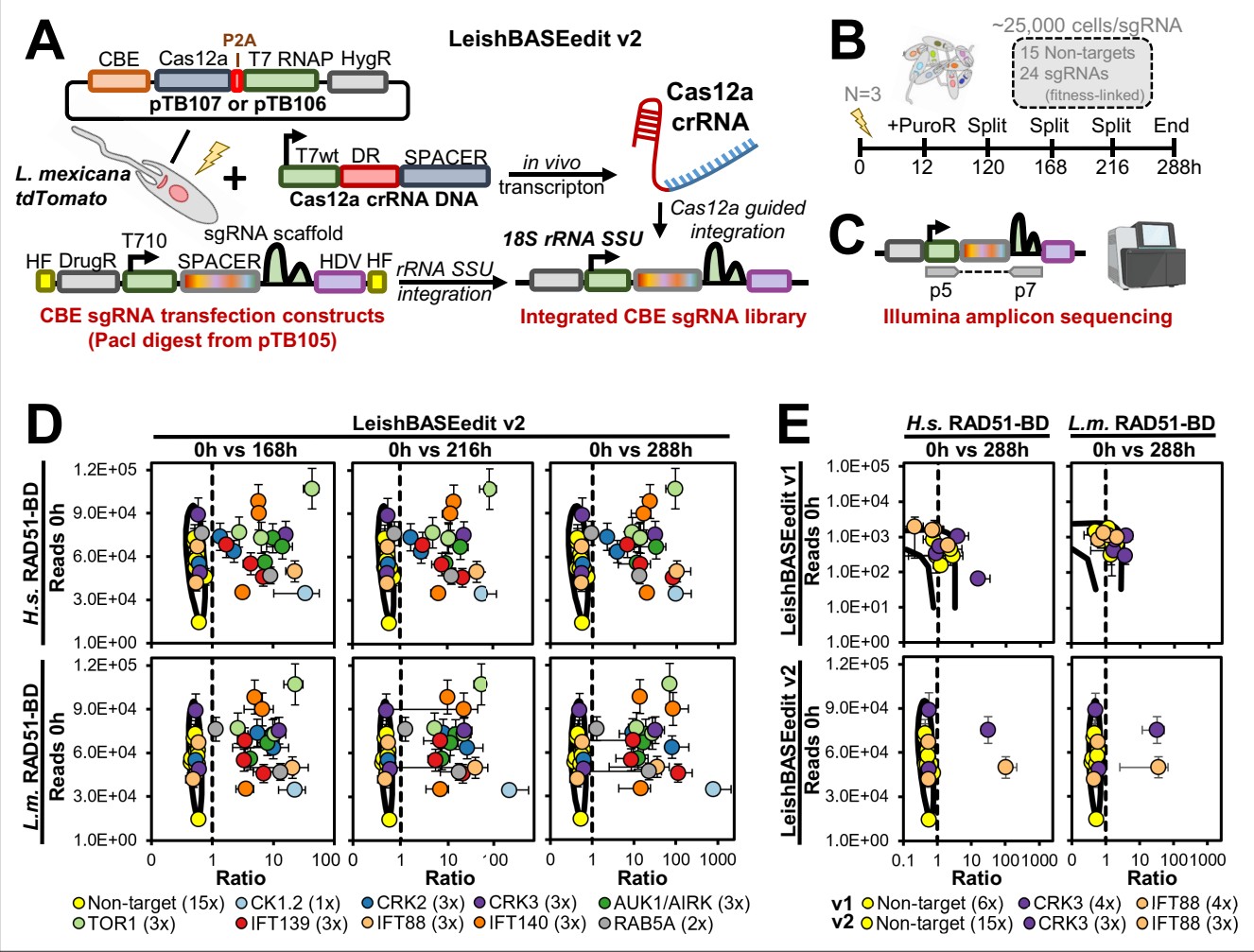

**Figure 5.** Small-scale loss-of-function screen. (**A**) Cas12a-mediated integration of the cytosine base editor (CBE) single-guide RNA (sgRNA) expression construct library involves co-transfecting a cell line expressing tdTomato and containing either pTB107 or pTB106 with two constructs: (1) a Cas12a crRNA template DNA and (2) a CBE sgRNA expression construct library. Integration is facilitated by Cas12a-mediated double-strand breaks (DSBs), as described in *Figure 3*. (**B**) The library, transfected in three independent replicates, contains 15 non-targeting sgRNAs and 24 sgRNAs targeting essential or fitness-associated genes. Plasmid library DNA and genomic DNA from transfected cells were extracted before transfection (0 hr) and after several subcultures (168, 216, and 288 hr post-transfection). (**C**) Samples underwent Illumina amplicon sequencing for downstream analysis. (**D and E**) For each sgRNA, the ratio of normalized counts between 0 hr and 168, 216, or 288 hr was calculated and plotted against raw read counts. Averages from triplicates are displayed with error bars indicating the standard deviation. A 0.99 confidence ellipse, calculated from non-targeting controls, highlights significant sgRNA depletion when data points fall outside the ellipse. CBEs used in the screen differ in the RAD51 single-stranded DNA-binding domain within the hyBE4max enzyme, derived from either *L. major* (pTB106: *L.m.* RAD51-BD) or *Homo sapiens* (pTB107: *H.s.* RAD51-BD). (**E**) For the comparison, data from *Engstler and Beneke, 2023* (LeishBASEedit v1) were re-analyzed and plotted alongside the newly generated data from this study (LeishBASEedit v2).

a control, our library also included 15 non-targeting sgRNA with random sequences (*Doench et al., 2016*).

The PacI-linearized library was then co-transfected with Cas12a crRNA-4 template DNA into *L. mexicana* carrying the pTB107 construct and the tdTomato expression cassette (*Figure 5A*). As shown earlier (*Figure 4*), these parasites did not reverse their phenotype when inducing a PF16 mutation. A second cell line with an alternative pTB107 construct, pTB106, containing the *Leishmania*-derived Rad51 single-stranded DNA-binding domain – previously shown to enhance cytosine base editing rates (*Engstler and Beneke, 2023*) – was also used (*Figure 5A*). Following transfection in three independent replicates, cells were selected for CBE sgRNA library integration at the 18S rRNA locus and cultured for 288 hr before undergoing Illumina amplicon sequencing (*Figure 5B and C*). The data analysis compared sgRNA ratios before and after transfection at various time points (*Figure 5D*).

As expected, sgRNAs targeting essential and fitness-associated genes were strongly depleted, in contrast to non-targeting control sgRNAs, which remained unchanged. Notably, only two IFT88, two CRK3, and one CRK2 sgRNAs showed no reduction in abundance, while all other sgRNAs moved significantly outside the 99% confidence interval of the non-targeting controls (*Figure 5D*). With a representation rate of ~25,000 cells/sgRNA, triplicate error margins were small. However, no substantial differences were observed between the two Rad51 variants used in the CBE system (*Figure 5D*). Lastly, we compared the data from our improved base editing method with results from a previous screen conducted using our first base editing method version (*Engstler and Beneke, 2023*). The comparison between method v1 and v2 revealed significant improvements: for one sgRNA targeting IFT88 and another targeting CRK3, the new method achieved ~10–20× stronger ratios with substantially reduced error margins for the non-targeting controls, enabling more accurate classification of fitness-associated genes (*Figure 5E*). Overall, these results demonstrate that our method has been significantly improved, enabling robust and scalable loss-of-function screens to identify essential and fitness-associated genes in the future. Notably, we have now completed the first genome-wide functional screens in *Leishmania* species using libraries containing up to 40,000 sgRNAs (Beneke lab, manuscript in preparation).

## Improved CBE sgRNA design to prioritize edits resulting only in STOP codons

The results above demonstrate that Cas12a crRNAs can be used to efficiently integrate CBE sgRNA expression cassettes and libraries into *Leishmania* species. This has no significant impact on parasite growth and enables high editing rates in all these tested species. The utility of fitness screens is also demonstrated. However, as we noted in our previous study (*Engstler and Beneke, 2023*), high editing rates do not necessarily lead to a strong knockdown of the targeted protein. Cells with a deleterious mutation can be simply outcompeted by cells with a non-deleterious mutation. This issue is particularly relevant when not all available cytosine nucleotides within the CBE sgRNA editing window result in a STOP codon after thymine conversion. To address this, we have therefore improved the scoring of previously designed CBE sgRNAs by implementing a C-score that determines the ratio of available cytosines to resulting STOP codons (*Figure 4—figure supplement 4*). This new scoring system has been uploaded to https://www.leishbaseedit.net and will be systematically tested in future studies, e.g., by using editing reporters. We also considered the position at which the STOP codon would occur and designed a scoring matrix that rates CBE sgRNAs higher when the STOP codon is introduced within positions 4–8 and lower when introduced within positions 11–12 (*Figure 4—figure supplement 4*). While our ratio scoring aims to increase the chance of successfully generating a functional mutation through STOP codon insertion, it is important to note that the introduction of a STOP codon is not the only efficient way to generate a functional mutant. As shown above, the introduction of a thymidine homo-polymer ('TTTTT') within the PF16 CDS produces the same motility phenotype observed when using gene replacement strategies for knockout generation (*Beneke et al., 2017*). Ultimately, this expands the possibilities for upcoming mutational screens in *Leishmania*.

## Co-expression of Cas12a and CBE enables a wide range of genetic manipulations

While these modifications to our sgRNA scoring tool represent a hypothetical but yet promising improvement in our method, the design of our triple expression construct (AsCas12a ultra, T7 RNAP, and Cas9 CBE) also offers opportunities for other advancements. For example, it enables hybrid approaches where classical CRISPR gene replacement and/or tagging methods can be combined with cytosine base editing experiments in one cell line. This is particularly interesting for targeting multiple genes simultaneously or dispersed multi-copy genes, and can also include the co-targeting of non-coding and coding genetic elements. Furthermore, since Cas12a has an intrinsic RNA cleavage activity, multiplexing of sgRNAs is possible and this expands the range of potential applications further as outlined above.

To explore the applicability of AsCas12a ultra for gene replacement strategies, such as the Leish-GEdit approach (*Beneke and Gluenz, 2019*; *Beneke and Gluenz, 2020*; *Beneke et al., 2017*), we devised two Cas12a crRNAs targeting the 5' and 3' UTR of the PF16 ORF (*Figure 6A*). Additionally, we designed 30 nt homology flanks adjacent to these crRNAs. We then PCR-amplified pTBlast and pTPuro

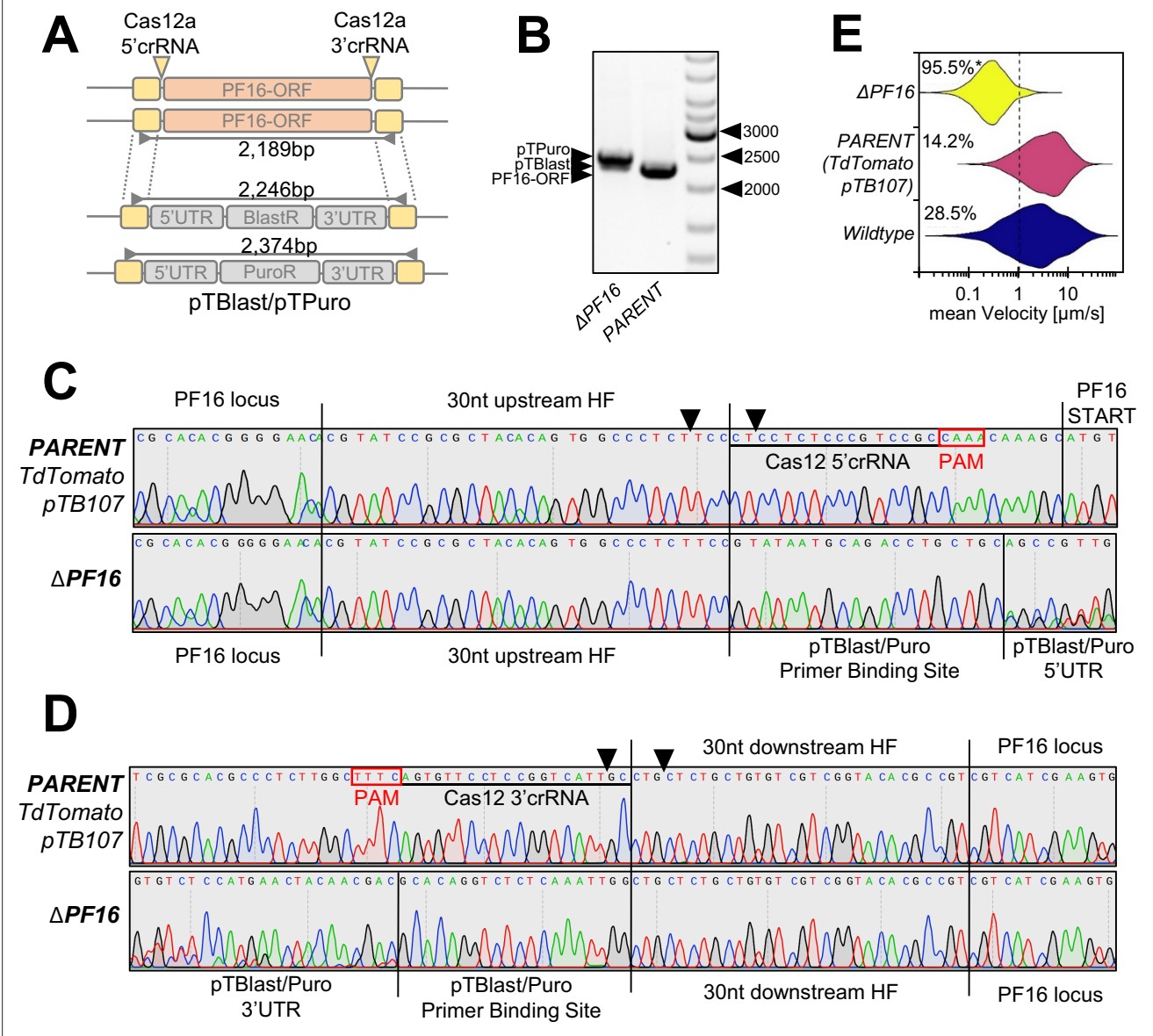

**Figure 6.** PF16 gene replacement by using CRISPR/AsCas12a ultra. (**A**) Map of the PF16 locus before and after targeting with Cas12a 5′ and 3′crRNA positions indicated. The PF16-ORF is replaced with a pTBlast and pTPuro cassette. Primers used for verification of the ORF replacement are highlighted with PCR amplicon length. (**B**) Visualization of PCR amplicons, showing expected products for pTPuro and pTBlast in the ΔPF16 lane and the amplicon of the PF16-ORF in the PARENT lane (*L. mexicana* tdTomato/pTB107 cell line). (**C and D**) Sanger sequencing trace plots of amplicons in (**B**), showing sequencing of the upstream region in (**C**) and downstream region in (**D**). Black arrows indicate Cas12a double-strand breaks (DSBs). (**E**) Violin plot of pooled replicates from motility tracked non-clonal populations, including the ΔPF16 and PARENT cell line. The total percentage of tracked cells showing a velocity of less than 1 μm/s is highlighted. Each population was analyzed using a Cramér-von Mises test to detect any shift in the population distribution toward lower speed. Percentages are marked with an asterisk when that shift was significant (*p>0.05). ORF, open-reading frame.

The online version of this article includes the following source data for figure 6:

**Source data 1.** Raw DNA images of *Figure 6B* with labels.

**Source data 2.** Raw DNA images of *Figure 6B* without labels.

cassettes from the LeishGEdit toolbox using primers containing these homology flanks and co-transfected them with Cas12a crRNA template DNA into *L. mexicana* parasites. These cells contained the pTB107 construct for co-expression of AsCas12a ultra, T7 RNAP, and CBE. Like the Cas9 LeishGEdit approach, transfected Cas12a crRNAs contained a T7 RNAP promoter for efficient in vivo transcription (*Figure 2B*). Following transfection, we then confirmed that the ORF was successfully replaced by

drug-resistance markers on both alleles and amplified the PF16 locus with two ORF spanning primers (*Figure 6A and B*). In addition, we performed Sanger sequencing over integration flanks to verify the integration of donor DNA fragments. This analysis confirmed the accurate integration of both donor DNA constructs (*Figure 6C and D*). Lastly, we performed again our motility analysis on these mutants, revealing the expected PF16 motility phenotype (*Figure 6E*). Overall, this demonstrates the versatility of our AsCas12a ultra, T7 RNAP, and CBE co-expression vector for a wide range of CRISPR applications in *Leishmania*.

## Conclusion

The *Leishmania* research community faces significant challenges in systematically assessing the function of all genes within the species. The absence of a functional NHEJ pathway complicates the process of obtaining null mutants, typically demanding the additional use of donor DNA, selection of edits associated with drug resistance, or the time-consuming isolation of clones. These complexities also apply to loss-of-function screens. While pooled CRISPR screens that rely solely on sgRNA expression are feasible to some extend for drug-resistance studies in *Leishmania* (*Queffeulou et al., 2024*), they are less likely to succeed in fitness-related phenotypic screens. While, bar-seq screens (*Baker et al., 2021*; *Beneke et al., 2019*; *Beneke and Gluenz, 2020*) and RNAi methods (*de Paiva et al., 2015*; *Lye et al., 2022*) offer potential solutions, they come with clear limitations. Most notably, the use of RNAi is restricted to *Leishmania* species within the *Viannia* subgenus, and bar-seq screens pose logistical challenges due to the necessity of generating thousands of individually created mutants. Furthermore, CRISPR gene replacement approaches, such as bar-seq methods, require multiple rounds of transfection when targeting dispersed multi-copy genes and are not suited to mutate interrupted tandemly arrayed coding and non-coding genes, which are abundantly present in the *Leishmania* genome.

Therefore, we recently introduced cytosine base editing as an alternative method for gene editing and large-scale screening in *Leishmania*, aiming to overcome these limitations. However, variations in editing efficiency, substantial reduction in parasite growth, competition between deleterious and non-deleterious mutations, and low transfection rates prompted further refinements of our method. Here, we present major improvements that collectively enhance our CBE method for *Leishmania* parasites.

In our final setup, we effectively employ a dual-construct system, where a stable cell line is transfected with a CBE sgRNA expression construct. The stable cell line expresses the hyBE4max CBE, AsCas12a ultra, and T7 RNAP, with CBE sgRNA expression driven by a T7 RNAP promoter variant (T7 T-10 GG). Alongside our updated CBE sgRNA design tool, which now prioritizes sgRNAs with a low cytosine to resulting STOP codon ratio, this system achieves high editing rates without affecting parasite growth. Compared to our initial CBE single-vector method, this dual-construct system has the disadvantage that it requires the generation of a stable cell line first. However, it offers the clear advantage that CBE sgRNA expression constructs can be efficiently integrated using AsCas12a-mediated DSBs. Through testing a series of Cas12a crRNAs, we identified a specific Cas12a crRNA capable of efficiently integrating CBE sgRNA expression cassettes into the 18S rRNA SSU safe harbor locus. CBE sgRNA expression cassettes can thereby be generated by amplification or digestion of the plasmid pTB105. In combination with the Lonza Nucleofector technology, this approach increases transfection rates by ~400-fold, yielding up to 1 transfectant per 70 transfected cells. This improved base editing method enabled the successful execution of a small-scale loss-of-function screen, confirming the essential and/or growth-associated roles of several kinases (CK1.2, CRK2, CRK3, AUK1/AIRK, and TOR1), as well as proteins related to intraflagellar transport (IFT88, IFT139, and IFT140) and endocytosis (Rab5A). Lastly, the co-expression of Cas12a and CBE provides strategic advantages, enabling the use of CRISPR gene replacement and base editing approaches in the same cell line. Moreover, Cas12a can be easily employed for protein tagging and multiplexing of sgRNAs, offering in the future a wide range of possible editing experiments in combination with base editing. Overall, our improved toolbox sets the stage for various gene editing applications in *Leishmania*, including genome-wide CBE screens, and we believe that these will make significant advancements in the field.

# Materials and methods

**Key resources table**

| Reagent type (species) or resource | Designation | Source or reference | Identifiers | Additional information |
|---|---|---|---|---|
| Cell line (*Leishmania mexicana*) | *L. mexicana* wildtype | Eva Gluenz laboratory | WHO strain MNYC/BZ/62/ M379 | Used TriTrypDB (release 59, *Aslett et al., 2010*) reference annotation: *L. mexicana* MHOMGT2001U1103 |
| Cell line (*Leishmania major*) | *L. major* wildtype | Eva Gluenz laboratory | Strain Friedlin | Used TriTrypDB (release 59, *Aslett et al., 2010*) reference annotation: *L. major* Friedlin |
| Cell line (*Leishmania donovani*) | *L. donovani* wildtype | Joachim Clos laboratory (*Decuypere et al., 2005*) | Strain BPK190 | Used TriTrypDB (release 59, *Aslett et al., 2010*) reference annotation: *L. donovani* BPK282A1 |
| Recombinant DNA reagent | pTB007-hyBE4max | *Engstler and Beneke, 2023* | | Contains CBE hyBE4max and T7 RNAP |
| Recombinant DNA reagent | pLdCH-hyBE4max | *Engstler and Beneke, 2023* | | Contains CBE hyBE4max and CBE sgRNA |
| Recombinant DNA reagent | pLdCH-hyBE4max-LmajDBD | *Engstler and Beneke, 2023* | | Contains *Leishmania*-optimized CBE hyBE4max and CBE sgRNA |
| Recombinant DNA reagent | pTB107 | This study | | Contains CBE hyBE4max, AsCas12a ultra, and T7 RNAP (available in versions with either a hygromycin or phleomycin-resistance marker) |
| Recombinant DNA reagent | pTB106 | This study | | Contains *Leishmania*-optimized CBE hyBE4max, AsCas12a ultra, and T7 RNAP (available in versions with either a hygromycin or phleomycin-resistance marker) |
| Recombinant DNA reagent | pTB102 | This study | | Contains T7 RNAP promoter-driven CBE sgRNA expression cassette without homology flanks |
| Recombinant DNA reagent | pTB104 | This study | | Contains T7 RNAP promoter-driven CBE sgRNA expression cassette with homology flanks for neomycin-resistance gene |
| Recombinant DNA reagent | pTB105 | This study | | Contains T7 RNAP promoter-driven CBE sgRNA expression cassette with homology flanks for 18S rRNA SSU locus |
| Software | TriTrypDB (release 59) | *Aslett et al., 2010* | https://tritrypdb.org/ tritrypdb/app | |
| Software | LeishBASEedit | This study | http://www.leishbaseedit. net/ | See description under 'Automated CBE guide design using LeishBASEedit' |
| Gene (*L. donovani* BPK282A1, *L. major* Friedlin, *L. mexicana* MHOMGT2001U1103) | PF16 | TriTrypDB (release 59) (*Aslett et al., 2010*) | LdBPK_201450.1, LmjF.20.1400, LmxM.20.1400 | |

## Cell culture

As described in *Engstler and Beneke, 2023*, promastigote-form *L. mexicana* (WHO strain MNYC/ BZ/62/M379), *L. major* Friedlin, and *L. donovani* (strain BPK190, *Decuypere et al., 2005*) were grown at 28°C in M199 medium (Life Technologies) supplemented with 2.2 g/L NaHCO$_3$, 0.0025% hemin, 0.1 mM adenine hemisulfate, 1.2 µg/mL 6-biopterin, 40 mM 4-(2-hydroxyethyl)piperazine-1-ethanesulfonic acid (HEPES) pH 7.4, and 20% FCS. Media were supplemented with the relevant selection drugs: 40 µg/ml hygromycin B, 40 µg/ml puromycin dihydrochloride, and 40 µg/ml G-418 disulfate. The identity of each *Leishmania* species and absence of mycoplasma contamination was confirmed previously (*Engstler and Beneke, 2023*). Doubling times were determined as described in *Engstler and Beneke, 2023*.

## Cas12a and CBE sgRNA design

For optimizing the CBE targeting of tdTomato and PF16, we selected sgRNAs from *Engstler and Beneke, 2023*, *Supplementary file 1*.

Cas12a crRNAs were designed using CCTop (*Labuhn et al., 2018*; *Stemmer et al., 2015*) with the 'PAM type' setting 'TTTN' and species specification 'Protists – Euglenozoa – *Leishmania tarentolae*'. For the Cas12a crRNA sequence search, we selected the 18S rRNA locus from *L. mexicana* (using an ONT-Illumina sequencing optimized MHOMGT2001U1103 genome annotation from *Beneke et al., 2022*) and selected homology flanks routinely used for integration into this locus (*Misslitz et al., 2000*; *Sörensen et al., 2003*). For designing Cas12a crRNAs that would target a potential 'landing-pad', we searched the neomycin-resistance marker for possible spacer sequences. All resulting Cas12a crRNAs were then checked for possible off-targets by using a local blast search against all *Leishmania* species currently available on TritrypDB (release 59, *Aslett et al., 2010*). This ensures that Cas12a crRNAs can be used for specific donor DNA integration in these species.

## Plasmid construction

All generated plasmids were subjected to whole plasmid sequencing at Plasmidsaurus and critical sites were additionally subject to Sanger sequencing.

Primers were ordered as standard desalted oligos at 25 nmole scale (Sigma). All oligo sequences can be found in *Supplementary file 1*. Plasmid maps are contained within *Supplementary file 2* and can also be downloaded from https://www.leishbaseedit.net/.

## Construction of CBE, Cas12a, and T7 RNAP co-expression plasmids

We synthesized the ORF of AsCas12a ultra (M537R/F870L mutation) (*Zhang et al., 2021*) with a fused SV40 nuclear localization signal (NLS), a 3xFLAG tag, a P2A self-cleaving peptide, and a c-myc NLS, consisting of six repeats. This c-myc NLS repeat has been previously shown to increase editing activity (*Gier et al., 2020*). We then amplified the synthesized Cas12a construct in a fusion PCR, adding parts of the T7 RNAP (ORF) and an intergenic region to the construct (using primer-pair 2051F/2052R and 2053F/2054R for amplification from pTB007-hyBE4max). Subsequently, this enabled cloning of the AsCas12a expression cassette into pTB007-hyBE4max using FseI and AflII. We termed the resulting plasmid pTB107, which allows for the co-expression of AsCas12a ultra, T7 RNAP, hyBE4max CBE, and hygromycin-resistance marker. To create a version of pTB107 that contained our previously pioneered *Leishmania*-derived version of the Rad51 single-stranded DNA-binding domain (ssDNA-DBD), we digested pLdCH-hyBE4max-LmajDBD (*Engstler and Beneke, 2023*) and pTB107 with AvrII and FseI to generate pTB106.

## Optimization of sgRNA expression promoters and construction of sgRNA expression plasmids and libraries

For generation of T7 RNAP promoter-driven CBE sgRNA expression plasmids, we amplified and cloned the sgRNA expression cassette from pLdCH into NsiI and MluI sites of a pPLOT-Puro plasmid (*Beneke et al., 2017*), thereby introducing the unmodified T7 RNAP promoter sequence (amplification using primer-pair 2128F-WT/2129R). To enable sgRNA cloning using BbsI sites, we then eliminated the BbsI site contained within the Actin 5'UTR of this plasmid, using primer-pair 2126F/2129R and PspOMI/MluI restriction sites. To avoid over-expression of the puromycin drug-resistance marker, we also eliminated the additional unmodified T7 RNAP promoter sequence contained in the plasmid backbone by amplifying the whole expression cassette (using primer-pair 2067F/2068R) and cloning it using PacI into the bacterial backbone of plasmid pTB007 (*Beneke et al., 2017*). We then cloned into this resulting plasmid T7 RNAP and rRNAP promoter variant cassettes, containing tdTomato targeting sgRNAs, by using again NsiI and MluI sites (primer-pairs 2014–2021 and 2025–2027). TdTomato-targeting sgRNA expression constructs were then transfected as episomes into *L. major* parasites that expressed a tdTomato reporter, T7 RNAP, and CBE. Following transfection recovery, the proportion of tdTomato-expressing cells was determined by FACS analysis and doubling times measured. Transfection and FACS analysis were thereby carried out as described previously (*Engstler and Beneke, 2023*). Following promoter testing, we then finalized our design and generated pTB102 with a T7 RNAP T10 GG promoter variant (using primer-pair 2128F-T10GG/2129R).

To include homology flanks in the pTB102 plasmid, neomycin-targeting or 18S rRNA-targeting regions were amplified and fused to pTB102 sgRNA expression cassettes using a fusion PCR approach (primer-pairs 2236F-2257R). Resulting amplicons were cloned into PacI sites, resulting in pTB104 (neomycin-targeting T7 RNAP expression plasmid) and pTB105 (18S rRNA-targeting T7 RNAP expression plasmid).

PF16 and tdTomato targeting CBE sgRNAs, as well as sgRNA libraries, were cloned into pTB102, pTB104, and/or pTB105 plasmids using BbsI sites as previously described (*Engstler and Beneke, 2023*).

## Preparation of donor DNA and Cas12a crRNA template DNA

To amplify the CBE sgRNA expression donor construct, 50 ng of plasmid template (pTB102, pTB104, or pTB105) containing a CBE targeting guide, 200 µM dNTPs, 0.5 µM each of forward and reverse primers, and 1 unit of Q5 polymerase (New England Biolabs) were mixed in 1× Q5 buffer to a final volume of 100 µl. Oligos used for amplification contained homology flanks of varying lengths. PCR steps were 30 s at 98°C, followed by 20 cycles of 10 s at 98°C, 10 s at 65°C, and 40 s at 72°C, concluding with a final elongation step of 5 min at 72°C. Alternatively, donor DNA was produced by digesting pTB104 and pTB105 plasmids with PacI. To confirm the presence of the expected product, 2 µl of this reaction was analyzed on a 0.8% agarose gel.

For the amplification of the Cas12a crRNA template DNA, we used a common forward primer, containing an unmodified T7 RNAP promoter sequence and an optimized Cas12a DR from *DeWeirdt et al., 2021*. For the PCR 200 µM dNTPs, 2 µM each of Cas12a forward primer and corresponding reverse primer, and 1 unit of Phusion Polymerase (Thermo Fisher) were mixed in 1× Phusion GC Buffer and 3% DMSO to a total volume of 50 µl. PCR steps were 30 s at 98°C followed by 35 cycles of 10 s at 98°C, 10 s at 65°C, and 10 s at 72° concluding with a final elongation step for 7 min at 72°C. Successful amplification was confirmed by running 2 µl of the reaction on a 2% agarose gel.

The remainder of the donor and crRNA reaction was combined, EtOH purified, resuspended in 50 µl ultrapure water, heat-sterilized at 95°C for 5 min, and then used for transfection.

## Testing Cas12a-mediated integration of mNG constructs

To test the Cas12a-mediated integration efficiency of donor DNA constructs with varying homology flank lengths, we generated a DNA construct containing mNG as a reporter and blasticidin as a resistance marker, as well as ~600 nt homology flanks targeting the 18S rRNA SSU locus (*Misslitz et al., 2000*; *Sörensen et al., 2003*). This construct was assembled by fusion PCR as previously described (*Engstler and Beneke, 2023*), using pPLOT-mNG-Blast (*Beneke et al., 2017*) as a PCR template. The generated fusion construct was then cloned into a blunted vector backbone, resulting in the plasmid pPLOT-SSU-HDR-mNG-Blast.

We then PCR-amplified as described above donor DNA from this plasmid with varying homology flank lengths (~600, 60, and 30 nt; *Supplementary file 1*). Resulting PCR amplicons and the pPLOT-SSU-HDR-mNG-Blast plasmid itself were then purified using ethanol precipitation and normalized to the same number of DNA molecules (3.6×10$^{12}$ molecules per transfection). This normalization was based on 10 µg of pPLOT-SSU-HDR-mNG-Blast plasmid. In addition, 10 µg of Cas12a crRNA template DNA was added to donor DNA as indicated (*Figure 2*). As described previously (*Engstler and Beneke, 2023*), DNA mixtures were then transfected using 5×10$^6$ cells per transfection and we determined the number of blasticidin-resistant clones by immediately diluting transfected populations and platting them on 96-well plates. In addition, 10 days post transfection undiluted blasticidin-resistant non-clonal populations were subjected to FACS analysis. To verify the correct integration of donor DNA, DNA was isolated from these populations as described (*Engstler and Beneke, 2023*; *Rotureau et al., 2005*) and amplified integration sites were submitted for Sanger sequencing at Eurofins Genomics (*Figure 2—figure supplement 1A and D*).

## High efficiency transfections

Transfections were generally carried out as described in *Engstler and Beneke, 2023*, transfecting ~5 × 10$^6$ cells in 1× Tb-BSF buffer (*Schumann Burkard et al., 2011*) using 2 mm cuvettes (BTX) with an Amaxa Nucleofector 2b (Lonza, program X-001).

For high efficiency transfections, we modified the protocol and compared side-by-side the transfection efficiency of the Basic Parasite Nucleofector Kit (Lonza) and Tb-BSF protocol (*Schumann Burkard et al., 2011*). $1 \times 10^8$ or $2.5 \times 10^8$ cells were collected, washed once in 1× Tb-BSF buffer, and resuspended in either 100 µl Lonza transfection reagent or 200 µl Tb-BSF buffer. 10 µg donor DNA and 10 µg Cas12a crRNA template DNA were diluted in either 20 µl (Lonza transfection) or 50 µl (Tb-BSF transfection) ultrapure water, heat-sterilized, and added to each transfection respectively (final transfection volume Tb-BSF 250 µl, Lonza 120 µl). Cells were transfected using transfection cuvettes supplied with the Basic Parasite Nucleofector Kit (Lonza transfection) or 2 mm BTX cuvettes (Tb-BSF transfection) with one pulse of the X-001 program on an Amaxa Nucleofector 2b (Lonza). For measuring transfection efficiencies, cells were selected and diluted on 96-well plates as described (*Engstler and Beneke, 2023*).

## Small-scale loss-of-function screen

For the small-scale loss-of-functions screen, the CBE sgRNA pTB105 library was transfected as described in the high efficiency transfections protocol above using $1 \times 10^8$ cells, 10 µg PacI-linearized library DNA, and 10 µg Cas12a crRNA template DNA. Six hours post transfection, non-clonal transfected library populations were selected with 40 µg/ml hygromycin B and 40 µg/ml puromycin dihydrochloride. The transfection efficiency was determined as previously described (*Engstler and Beneke, 2023*), yielding approximately 1 transfectant per 100 transfected cells. This corresponded to a representation rate of ~25,000 cells per sgRNA within the transfected library. Cultures were maintained for up to 288 hr with sub-cultures whenever the cell density reached $1 \times 10^7$ cells/ml. At each time point, DNA was isolated from at least $1 \times 10^7$ cells, as described previously (*Engstler and Beneke, 2023*; *Rotureau et al., 2005*). Then, 1 µg DNA from each time point and 10 ng DNA from plasmid libraries were amplified using FastGene Optima HotStart Ready Mix (NIPPON; contains all ingredients for PCR). Amplification employed standard desalted p5 and p7 primers (Sigma, *Supplementary file 1*), containing inline and i5/i7 barcodes for multiplexing, as well as adapters for Illumina sequencing. Barcodes had a Hamming distance of at least 4 nt. To avoid over-amplification, samples isolated from transfected populations were amplified using 26 PCR cycles, while plasmid samples were amplified using only 16 PCR cycles (ideal number of PCR cycles was assessed by testing 12, 14, 16, 18, and 20 PCR cycles for plasmid samples and 22, 24, 26, 28, 30, and 32 PCR cycles for samples isolated from transfected cells). Amplicons were pooled in equal ratios, size-selected using SPRIselect beads (Beckman Coulter), and send to Novogene GmbH for partial-lane Illumina sequencing (150 bp paired-end sequencing).

For analysis reads were de-multiplexed twice: first using bcl2fastq (Illumina) with i5 and i7 indices, and then with cutadapt (*Martin, 2011*) using inline barcodes (with a 0.15 error rate). This double de-multiplexing minimized index hopping. De-multiplexed forward reads were counted and normalized using MAGeCK (*Li et al., 2014*). The ratio of normalized reads between plasmid (0 hr) and each time point post transfection was calculated. A 0.99 confidence ellipse was generated based on non-targeting control values alone (*Supplementary file 4*).

## Targeting of PF16 with an improved CBE system

For testing the editing efficiency of our improved CBE system, we amplified donor DNA from the pTB102 plasmid, containing the sgRNA PF16-3 from *Engstler and Beneke, 2023*. As indicated (*Figure 4*), donor DNAs were mixed with respective Cas12a crRNA DNA templates and transfected into *Leishmania* parasites. Following selection and the recovery of parasites, non-clonal mutant populations were tracked in duplicates or triplicates and their mean velocity determined as described before (*Engstler and Beneke, 2023*). Editing rates were measured by amplifying the PF16 target locus and by subjecting PCR amplicons to Sanger sequencing (*Engstler and Beneke, 2023*).

## ONT and Illumina sequencing analysis of CBE PF16 non-clonal mutant population

DNA from *L. mexicana* CBE PF16 non-clonal mutant populations was isolated as previously described (*Engstler and Beneke, 2023*; *Rotureau et al., 2005*) 16 days post transfection and submitted for ONT and Illumina sequencing at Plasmidsaurus (service: 'Big Hybrid ONT+Illumina'). Obtained raw fastq files from ONT and Illumina sequencing were then assembled using the Burrows-Wheeler Aligner (*Li*

*and Durbin, 2009*) using two customized *L. mexicana* genomes of the MHOMGT2001U1103 annotation. In the first genome, we duplicated chromosome 27 and modified the 18S rRNA SSU locus of one chromosome 27 copy to include the tdTomato expression cassette (*Engstler and Beneke, 2023*; *Figure 4—figure supplement 1B*), while the other copy contained the CBE sgRNA expression construct. In the second genome, we instead modified chromosome 27 to contain both expression cassettes consecutively (*Figure 4—figure supplement 1C*). Following alignments, we then used samtools (*Li et al., 2009*) for sorting and indexing of bam files and viewed our analysis using the IGV genome browser (*Robinson et al., 2011*). ONT and Illumina fastq reads are available at the European Nucleotide Archive (ENA) (Accession number: PRJEB83088).

For the standard nucleotide BLAST analysis (*Altschul et al., 1990*), ONT fastq reads were queried against the CDS of the tdTomato reporter gene and the sgRNA expression cassette. The cassette included the optimized T7 RNAP promoter (T7 T10 GG), PF16-3 targeting spacer, and Cas9 sgRNA scaffold sequence. Extracted contigs were then mapped to the 18S rRNA integration locus and adjacent genomic sites using Clustal Omega (*Sievers et al., 2011*).

## Optimized CBE sgRNA scoring

To determine the likelihood of successful gene editing, we adjusted the sgRNA ranking system. New sgRNA ranking scores were calculated based on various parameters, including the relative likelihood of generating a STOP codon from a cytosine base, the total number of possible STOP codons, the location of cytosines within the editing window, and the position of the guide's target within the CDS of the gene. To generate the overall score, we weighted the contribution of each of these scores, with the likelihood of generating a STOP codon having four times more contribution than the other scores. We then normalized the total score on a scale of 0–100, where 0 represents the best score, and 100 represents the worst score.

The updated data sets and code can be accessed at http://www.leishbaseedit.net/ and https://github.com/ElisabethMeiser/Collaboration_Beneke_Meiser, copy archived at *Meiser, 2024* respectively.

## Cas12a-mediated gene replacement for generation of knockout cell lines

For the Cas12a-mediated replacement of the PF16 CDS, we adapted the LeishGEdit approach (*Beneke and Gluenz, 2019*; *Beneke and Gluenz, 2020*; *Beneke et al., 2017*) and PCR-amplified two Cas12a crRNAs targeting the 5' and 3' UTR of the PF16 ORF (crRNA design as described above). In addition, we PCR-amplified pTBlast and pTPuro cassettes from the LeishGEdit toolbox with primers containing 30 nt homology flanks adjacent to these crRNAs. Both types of PCR were carried out as described above and we then co-transfected all resulting products into *L. mexicana* cells containing the pTB107 plasmid. Successful gene replacement in obtained non-clonal mutant populations was confirmed by amplifying and Sanger sequencing the PF16 locus with two ORF spanning primers.

## Acknowledgements

We thank all members of the Alsheimer, Engstler, Janzen, and Kramer group for helpful discussions and support. In addition, we thank Markus Engstler for providing research resources and helpful comments on the manuscript. We thank Ger van Zandbergen from the Paul-Ehrlich-Institut for his support in acquiring funding for JA and FL, who were supported by a Flex fund within the LOEWE Center DRUID (Project D3, B3). TB and NHM were supported by the DFG (project 532631727). Additionally, TB was supported by an EMBO Postdoctoral Fellowship (ALTF 727-2021) and Marie Skłodowska-Curie Actions Postdoctoral Fellowship (101064428 – LeishMOM).

## Additional information

### Funding

| Funder | Grant reference number | Author |
|---|---|---|
| European Molecular Biology Organization | ALTF 727-2021 | Tom Beneke |
| HORIZON EUROPE Marie Sklodowska-Curie Actions | 101064428 - LeishMOM | Tom Beneke |
| Deutsche Forschungsgemeinschaft | 532631727 | Tom Beneke |
| LOEWE Center DRUID | Project D3 and B3 | Tom Beneke |

The funders had no role in study design, data collection and interpretation, or the decision to submit the work for publication.

### Author contributions

Nicole Herrmann May, Data curation, Formal analysis, Validation, Investigation, Visualization, Writing – review and editing; Anh Cao, Annika Schmid, Investigation; Fabian Link, Jorge Arias-del-Angel, Validation, Investigation, Writing – review and editing; Elisabeth Meiser, Software, Writing – review and editing; Tom Beneke, Conceptualization, Resources, Data curation, Software, Formal analysis, Supervision, Funding acquisition, Validation, Investigation, Visualization, Methodology, Writing – original draft, Project administration, Writing – review and editing

### Author ORCIDs

Fabian Link ⓘ https://orcid.org/0000-0002-9828-2012
Tom Beneke ⓘ https://orcid.org/0000-0001-9117-2649

Reviewer #1 (Public review): https://doi.org/10.7554/eLife.97437.3.sa1
Reviewer #2 (Public review): https://doi.org/10.7554/eLife.97437.3.sa2
Reviewer #3 (Public review): https://doi.org/10.7554/eLife.97437.3.sa3
Author response https://doi.org/10.7554/eLife.97437.3.sa4

## Additional files

### Supplementary files

Supplementary file 1. Primers used in this study.

Supplementary file 2. Plasmid maps. GenBank files of pTB102, pTB104, pTB105, pTB106, and pTB107.

Supplementary file 3. A protocol for AsCas12a-mediated transfection of cytosine base editor (CBE) single-guide RNA (sgRNA) expression cassettes. Step-by-step protocol includes guidelines for PCR amplification of donor DNA and Cas12a crRNA template DNA, as well as preparations for transfection.

Supplementary file 4. Analysis of small-scale loss-of-function screen. The analysis file contains raw and normalized amplicon sequencing counts for each single-guide RNA (sgRNA) subjected to the screen, along with the ratio analysis shown in *Figure 5*.

MDAR checklist

Source data 1. Raw DNA images of *Figure 2—figure supplement 1B and E*. Details are described in *Figure 2—figure supplement 1*.

Source data 2. Raw DNA images of *Figure 6B*. Details are described in *Figure 6*.

Source code 1. Novel scoring for cytosine base editor (CBE) single-guide RNA (sgRNA) sorting. Source code for CBE sgRNA scoring described above. The updated data sets can be accessed at https://www.leishbaseedit.net/. The source code for the guide scoring has been uploaded to https://github.com/ElisabethMeiser/Collaboration_Beneke_Meiser, copy archived at *Meiser, 2024*.

## Data availability

All data generated or analyzed during this study are included in the manuscript and supplementary files. LeishBASEedit is an open-source primer design tool accessible at https://www.leishbaseedit.net. The code for scoring cytosine base editor guide sequences is available under https://github.com/ElisabethMeiser/Collaboration_Beneke_Meiser (copy archived at *Meiser, 2024*). ONT and Illumina fastq reads are available at ENA (Accession number: PRJEB83088). Raw imaging files for DNA gels are included in the supplementary files.

The following dataset was generated:

| Author(s) | Year | Dataset title | Dataset URL | Database and Identifier |
|-----------|------|---------------|-------------|-------------------------|
| Beneke T | 2024 | Nanopore and Illumina sequencing of CBE mutated Leishmania | https://www.ebi.ac.uk/ena/browser/view/PRJEB83088 | ArrayExpress, PRJEB83088 |

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
