## [Editor Report · eLife Assessment]

This **important** article describes a meticulously-developed improved strategy for generation of functionally null mutants in Leishmania spp. via cytosine base editing, with reduced background toxicity and enhanced efficiency relative to a previously-described method. The authors show use of the strategy in a small-scale loss-of-function screen, providing **compelling** evidence that large-scale screens will be possible. The newly developed tools will be of great interest to researchers working with Leishmania and beyond.

---

## [Referee Report · Reviewer #1 (Public review)]

While CRISPR/Cas technology has greatly facilitated the ability to perform precise genome edits in Leishmania spp., the lack of a non-homologous DNA end-joining (NHEJ) pathway in Leishmania has prevented researchers from performing large-scale Cas-based perturbation screens. With the introduction of base editing technology to the Leishmania field, the Beneke lab has begun to address this challenge (Engstler and Beneke, 2023). In this study, the authors build on their previously published protocols and develop a strategy that:

a) allows for very high editing efficiency. The cell editing frequency of 1 edit per 70 cells reported in this study represents a 400-fold improvement over the previously published protocol,

b) reduces the negative effects of high sgRNA levels on parasite growth by using a weaker T7 promoter to drive sgRNA transcription.

The combination of these two improvements should open the door to exciting large-scale screens and thus be of great interest to researchers working with Leishmania and beyond.

The authors did a great job responding to our concerns and we have no doubt that the technology established here, will be very useful for the Leishmania research community and beyond.

---

## [Referee Report · Reviewer #2 (Public review)]

Previously, the authors published a Leishmania cytosine base editor (CBE) genetic tool that enables the generation of functionally null mutants. This works by utilising a CAS9-cytidine deaminase variant that is targeted to a genetic locus by a small guide RNA (sgRNA) and causes a cytosine to thymine conversion. This has the potential to generate a premature stop codon and therefore a loss of function mutant.

CBE has advantages over existing CAS-based knockout tools because it allows the targeting of multicopy gene families and, potentially, the easier generation of pooled loss of function mutants in complex population experiments. Although successful, the first generation of this genetic tool had several limitations that may have prevented its wider adoption, especially in complex genome-wide screens. These include nonspecific toxicity of the sgRNAs, low transfection efficiencies, low editing efficiencies, a proportion of transfectants that express multiple different sgRNAs, and insufficient effectivity in some Leishmania species.

Here, the authors set out to systematically solve each of these limitations. By trialling different transfection conditions and different CAS12a cut sites to promote sgRNA expression cassette integration, they increase the transfection efficiency 400-fold and ensure that only a single sgRNA expression cassette integrates that edits with high efficiencies. By trialling different T7 promoters, they significantly reduce the non-specific toxicity of sgRNA expression whilst retaining high editing efficiencies in several Leishmania species (Leishmania major, L. mexicana and L. donovani). By improving the sgRNA design, the authors predict that null mutants will be more efficiently produced after editing. They validate this tool in a small-scale loss of function screen incorporating essential and non-essential genes, identifying the expected growth phenotypes.

This tool will find adoption for producing null mutants of single-copy genes, multicopy gene families, and genome-wide mutational analyses.

This is an impressive and thorough study that significantly improves the previous iteration of the CBE. The approach is careful and systematic and reflects the authors excellent experience developing CRISPR tools. The quality of data and analysis is high and data are clearly presented.

---

## [Referee Report · Reviewer #3 (Public review)]

Genetic manipulation of Leishmania has some challenges, including some limitations in the DNA repair strategies that are present in the organism and the absence of RNA interference in many species. The senior author has contributed significantly to expanding the available routes towards Leishmania genetic manipulation by developing and adapting CRISPR-Cas9 tools to allow gene manipulation via DNA double strand break repair and, more recently, base modification. This work seeks to improve on some limitations in the tools previously described for the latter approach of base modification leading to base change.

The work in the paper is meticulously described, with solid evidence for the improvements that are claimed: Fig.1 clearly describes reduced impairment in growth of parasites expressing sgRNAs via changes in promoters; Figs.2 and 3 compellingly document the usefulness of using AsCas12a for integration after transformation; Figs.1 and 4 demonstrate the capacity of the combined modifications to efficiently edit a gene in three different Leishmania species; and Fig. 5 shows that this approach can be conducted at scale, providing a means of assessing the fitness of mutant pools. There is little doubt these new tools will be adopted by the Leishmania community, adding to the growing arsenal of approaches for genetic manipulation.

Two weaknesses suggested in the initial submission have been completely addressed.

---

## [Author Response]

The following is the authors’ response to the original reviews.

**Public Reviews:**

**Reviewer #1 (Public Review):**
While CRISPR/Cas technology has greatly facilitated the ability to perform precise genome edits in Leishmania spp., the lack of a non-homologous DNA end-joining (NHEJ) pathway in Leishmania has prevented researchers from performing large-scale Cas-based perturbation screens. With the introduction of base editing technology to the Leishmania field, the Beneke lab has begun to address this challenge (Engstler and Beneke, 2023).In this study, the authors build on their previously published protocols and develop a strategy that:(1) allows for very high editing efficiency. The cell editing frequency of 1 edit per 70 cells reported in this study represents a 400-fold improvement over the previously published protocol,(2) reduces the negative effects of high sgRNA levels on parasite growth by using a weaker T7 promoter to drive sgRNA transcription.The combination of these two improvements should open the door to exciting large-scale screens and thus be of great interest to researchers working with Leishmania and beyond.

We thank reviewer #1 for these encouraging comments.

**Reviewer #2 (Public Review):**
Summary:Previously, the authors published a Leishmania cytosine base editor (CBE) genetic tool that enables the generation of functionally null mutants. This works by utilising a CAS9-cytidine deaminase variant that is targeted to a genetic locus by a small guide RNA (sgRNA) and causes cytosine to thymine conversion. This has the potential to generate a premature stop codon and therefore a loss of function mutant.CBE has advantages over existing CAS-based knockout tools because it allows the targeting of multicopy gene families and, potentially, the easier generation of pooled loss of function mutants in complex population experiments. Although successful, the first generation of this genetic tool had several limitations that may have prevented its wider adoption, especially in complex genome-wide screens. These include nonspecific toxicity of the sgRNAs, low transfection efficiencies, low editing efficiencies, a proportion of transfectants that express multiple different sgRNAs, and insufficient effectivity in some Leishmania species.Here, the authors set out to systematically solve each of these limitations. By trialling different transfection conditions and different CAS12a cut sites to promote sgRNA expression cassette integration, they increase the transfection efficiency 400-fold and ensure that only a single sgRNA expression cassette integrates that edits with high efficiencies. By trialling different T7 promoters, they significantly reduce the non-specific toxicity of sgRNA expression whilst retaining high editing efficiencies in several Leishmania species (Leishmania major, L. mexicana and L. donovani). By improving the sgRNA design, the authors predict that null mutants will be more efficiently produced after editing.This tool will find adoption for producing null mutants of single-copy genes, multicopy gene families, and potentially genome-wide mutational analyses.Strengths:This is an impressive and thorough study that significantly improves the previous iteration of the CBE. The approach is careful and systematic and reflects the authors' excellent experience developing CRISPR tools. The quality of data and analysis is high and data are clearly presented.Weaknesses:Figure 4 shows that editing of PF16 is 'reversed' between day 6 and day 16 in L. mexicana WTpTB107 cells. The authors reasonably conclude that in drug-selected cells there is a mixed population of edited and non-edited cells, possibly due to mis-integration of the sgRNA expression construct, and non-edited cells outcompete edited cells due to a growth defect in PF16 loss of function mutants. However, this suggests that the CBE tool will not work well for producing mutants with strong fitness phenotypes without incorporating a limiting dilution cloning step (at least in L. mexicana and quite possibly other Leishmania species). Furthermore, it suggests it will not be possible to incorporate genes associated with a growth defect into a pooled drop-out screen as described in the paper. This issue is not well explored in the paper and the authors have not validated their tool on a gene associated with a severe growth defect, or shown that their tool works in a mixed population setting.

We would like to thank reviewer #2 for this helpful comment and valid point. We have now included a small-scale loss-of-function screen in *L. mexicana*, targeting nine known essential genes with 24 CBE sgRNAs and 15 non-targeting control sgRNAs. This approach successfully detected all known included growth-associated phenotypes in a pooled screening format. This experiment is now shown in Figure 5 and described in section “Detection of fitness-associated phenotypes in a pooled loss-of-function screen”.

In addition, we would like to re-iterate our initial public response to this comment. We believe that escapes or reversals of mutant phenotypes can be observed also with other genetic tools used for loss-of-function screening, including lentiviral CRISPR approaches in mammalian systems and RNAi in *Trypanosoma brucei* (e.g. Ariyanayagam et al., 2005 and Schlecker et al., 2005). Notably, in lentiviral delivered CRISPR screens, sgRNA expression cassettes are integrated in random places within the genome and multiple cassettes can be integrated depending on the viral titre. In these type of screens, cells can escape phenotypes through various mechanisms, such as promoter silencing or selection of non-deleterious mutations. Additionally, not every CRISPR guide is efficient in generating a mutant phenotype, and RNAi constructs can also vary in their effectiveness. Despite these challenges, genome-wide loss-of-function screens have been successfully carried out in mammalian cells and Trypanosoma parasites. Therefore, we believe that the observed escape of one mutant phenotype does not preclude the detection of growth-associated or other phenotypes in pooled screens. Moreover, we did not observe a reversal of the mutant phenotype in *L. mexicana*, *L. donovani*, and *L. major* parasites expressing tdTomato from an expression cassette integrated into the 18S rRNA SSU locus (Figure 4). Our now included small scale fitness screen (Figure 5) confirms these assumptions and shows that we can detect “strong” growth associated phenotypes. We would also like to point out that we have recently successfully conducted several genome-wide loss-of-function screens in vivo and in vitro, ultimately confirming the feasibility of this type of screen on a genome-wide scale (manuscript in preparation).

We have included a discussion of these points under section “Integration of CBE sgRNA expression cassettes via AsCas12a ultra-introduced DSBs increase editing rates” and section “Detection of fitness-associated phenotypes in a pooled loss-of-function screen” in our revised manuscript.

Although welcome, the improvements to the crRNA CBE design tool are hypothetical and untested.

We agree that the improvements to the CBE sgRNA design are currently hypothetical. We plan to systematically test our guide design principles in future studies. Since this will require testing hundreds of guides to draw robust conclusions, we believe that this aspect is beyond the scope of the current study. In section “Improved CBE sgRNA design to prioritize edits resulting only in STOP codons” of our revised manuscript we now discuss these future plans.

The Sanger and Oxford Nanopore Technology analyses on integration sites of the sgRNA expression cassette integration will not detect the mis-integration of the sgRNA expression construct into an entirely different locus.

We have now re-analysed our ONT data and have extracted all ONT contigs that match the CBE sgRNA expression cassette. All extracted contigs align to the 18S rRNA SSU locus, showing integration of the cassette into this locus. It is important to note that here a population was sequenced and not a clone. Despite this, no contigs could be found that would link the CBE sgRNA expression cassettes to another locus. This is now shown in Figure 4 S2 and described in section “Cas12a-mediated DSB ensures the integration of one CBE sgRNA per *L. mexicana* transfectant”.

**Reviewer #3 (Public Review):**
Genetic manipulation of Leishmania has some challenges, including some limitations in the DNA repair strategies that are present in the organism and the absence of RNA interference in many species. The senior author has contributed significantly to expanding the available routes towards Leishmania genetic manipulation by developing and adapting CRISPR-Cas9 tools to allow gene manipulation via DNA double-strand break repair and, more recently, base modification. This work seeks to improve on some limitations in the tools previously described for the latter approach of base modification leading to base change.The work in the paper is meticulously described, with solid evidence for most of the improvements that are claimed: Figure1 clearly describes reduced impairment in the growth of parasites expressing sgRNAs via changes in promoters; Figures 2 and 3 compellingly document the usefulness of using AsCas12a for integration after transformation; and Figures 1 and 4 demonstrate the capacity of the combined modifications to efficiently edit a gene in three different Leishmania species. There is little doubt these new tools will be adopted by the Leishmania community, adding to the growing arsenal of approaches for genetic manipulation.There are two weaknesses the authors may wish to address, one smaller and one larger.(1) The main advance claimed here is in this section title: 'Integration of CBE sgRNA expression cassettes via AsCas12a ultra-introduced DSBs increase editing rates', with the evidence for this presented in Figure 4. It is hard work in the submission to discern what direct evidence there is for editing rates being improved relative to earlier, Cas9-based approaches. Did they directly compare the editing by the new and old approach? If not, can they more clearly explain how they are able to make this claim, either by adding text or a new figure? A side-by-side comparison would emphasise the advance of the new approach more clearly.

We would like to thank reviewer #3 for this helpful comment. We have directly compared our improved method to our previous base editing method in Figures 1E and 4, demonstrating higher editing rates in a much shorter time. Especially the L. major panel in Figure 4B shows that in a direct comparison between the previously published (Engstler and Beneke, eLife 2023) and our here presented new system, editing can be only observed with the version presented here. However, to clarify the improvements we made, we compare now data from our previous screen done in Engstler and Beneke, eLife 2023 with a loss-of-function screen carried out with our updated method (see Figure 5 and section “Detection of fitness-associated phenotypes in a pooled loss-of-function screen”).

In addition, we also feel that our title might have been misleading in a sense that we claim that Cas12a editing is more efficient than other Cas9 based approaches, which is something that we don’t want to state here. Given that we have now included a small scale CRISPR screen and given that we generally show improved base editing compared to our previous method (improved in terms of less toxicity, more editing in shorter time, higher transfection rates and less species specific variation), we have rephrased our title to: “Improved base editing and functional screening in Leishmania via co-expression of the AsCas12a ultra, a T7 RNA Polymerase, and a cytosine base editor”.

(2) The ultimate, stated goal of this work is (abstract) to 'enable a variety of loss-of-function screens', as the older approach had some limitations. This goal is not tested for the new tools that have been developed here; the experiment in Figure 5 merely shows that they can, not unexpectedly, make a gene mutant, which was already possible with available tools. Thus, to what extent is this paper describing a step forward? Why have the authors not run an experiment - even the same one that was described previously in Engstler and Beneke (2023) - to show that the new approach improves on previous tools in such a screen, either in scale or accuracy?

We have now included a small-scale loss-of-function screen in *L. mexicana*, targeting nine known essential genes with 24 CBE sgRNAs and 15 non-targeting control sgRNAs. This approach successfully detected all known included growth-associated phenotypes in a pooled screening format. This experiment is now shown in Figure 5 and described in section “Detection of fitness-associated phenotypes in a pooled loss-of-function screen”. We believe that this underscores our claims made here and believe therefore that our updated toolbox will indeed enable a variety of loss-of-function screens.

As pointed out in the comment to reviewer #2, we have recently successfully conducted several genome-wide loss-of-function screens in vivo and in vitro, ultimately confirming the feasibility of this type of screen on a genome-wide scale (manuscript in preparation). Without the improvements presented here, such as the higher transfection and base editing rates, these genome-wide screens could have not been carried out.

**Recommendations for the authors:**

**Reviewer #1 (Recommendations For The Authors):**
I would like to compliment Tom Beneke and his lab on their continued efforts to develop tools to facilitate genome editing in Leishmania.I have no doubt that the toolkit presented in this study will be very useful for the community. The submitted paper is very well written and contains all the necessary controls to support the author's claims. There is only one point that left me a bit concerned if this strategy is to be used for large-scale screens, and that is the potential for integration of multiple sgRNA expression cassettes in a single cell.

We would like to thank reviewer 1 for helpful comments. We have addressed the major concern raised by including a small-scale loss-of-function screen in our revised manuscript. By targeting nine known essential genes with 24 CBE sgRNAs and 15 non-targeting control sgRNAs, this approach successfully detected growth-associated phenotypes in a pooled format (see section “Detection of fitness-associated phenotypes in a pooled loss-of-function screen” and Figure 5). Regarding the point of multiple sgRNA expression cassette integration, please see the next comment below.

Major points:Integration of multiple sgRNA expression cassettes:While Illumina-based gDNA-seq is well suited to determine changes in ploidy, I don't think it is sensitive enough to draw conclusions about possible double integration in a small percentage of cells. In fact, the data shown in Figure 4 S1D show a normalized coverage >1.5 for sgRNA cassette and NeoR, suggesting that they may have integrated >1 times in some cells.

To verify that the integration of the CBE sgRNA expression cassette is specific, we have re-analysed our ONT results and confirmed that only ONT contigs can be detected that link the CBE sgRNA expression to the 18S rRNA locus. No other integration sites can be found. We also do not detect any contigs containing multiple CBE sgRNA expression cassettes. This is now shown in Figure 4 S2 and described in section “Cas12a-mediated DSB ensures the integration of one CBE sgRNA per *L. mexicana* transfectant”.

Nevertheless, it is a valid concern that the sequencing depth is not sufficient to detect small percentage of cells that have integrated the CBE sgRNA expression multiple times. However, in this case we also like to make the point that this small percentage of cells within a screen is likely to be not relevant and we therefore now added a small scale pooled loss-of-function screen, targeting essential genes, to the manuscript (see new Figure 5) to proof our claim. If the integration of multiple sgRNAs into one cell would have any measurable combinatorial effect, the non-targeting controls in our screen would have been depleted as well. However, there is no detectable difference between all 15 included controls in our small-scale screen.

We have addressed all points in sections “Cas12a-mediated DSB ensures the integration of one CBE sgRNA per L. mexicana transfectant“ and “Detection of fitness-associated phenotypes in a pooled loss-of-function screen”.

To avoid double integration, wouldn't it be easiest to just create an allele-specific "landing pad" on one chromosome? I believe that a double integration rate of ~20% could severely complicate the analysis of any large-scale screen later on.

We thank the reviewer for this suggestion but we have tried to use an allele-specific "landing pad" and described this already in our first manuscript version (see section “DSBs introduced by AsCas12a ultra increase integration rates of donor DNA constructs”). Specifically, we integrated CBE sgRNA expression cassettes into the neomycin resistance marker contained in the tdTomato expression cassette (Figure 2 S1D, Cas12a crRNA-5 and 6) but this resulted in lower transfection rates (Figure 2F: crRNA-5 1 in ~47,000; crRNA-6 1 in ~32,000) then when using a Cas12a crRNA that targets the 18S rRNA locus directly (Figure 2F: crRNA-4 1 in ~2,000). As we believe a high transfection rate is key for pooled large-scale screens, we therefore pursued further experiments with crRNA-4. However, since a different crRNA can be easily selected for our tool, simply by just changing the Cas12a crRNA during transfection, users can chose a different integration site or other “landing pads” if they want to. We have updated section “Cas12a-mediated DSB ensures the integration of one CBE sgRNA per L. mexicana transfectant” to clarify these details.

Also, it is not clear to me how the integration of tdTomato could affect the integration of the sgRNA expression cassette 400 bp downstream.

As said above, our ONT data clearly shows that we can only see integration into one locus (Figure 4 S1 and S2). Given that the recognition site of crRNA-4 is contained in the homology flank used to integrate tdTomato into the 18S rRNA locus, this may contribute to the effect we observe. But since the homology sequences match the original sequences within the locus, the reasons to why this affects integration of the CBE sgRNA expression cassettes remain also elusive to us. We try to discuss this better now in the section “Cas12a-mediated DSB ensures the integration of one CBE sgRNA per L. mexicana transfectant”.

Data accessibility:The Illumina and ONT data should be made publicly available.

ONT and Illumina fastq reads are now available at the European Nucleotide Archive (ENA Accession Number: PRJEB83088)

Minor point:Line 30: It would be easier for readers if the authors could briefly explain what bar-seq is.

We have added more details:[…] and bar-seq screens, which involve individually deleting, barcoding, and pooling mutants for analysis, have facilitated […].

Lines 114, 120: I think the authors are referring to Figures 1E and F, not Figures 2E and F.

Many thanks for picking this up, we have corrected the Figure reference.

**Reviewer #2 (Recommendations For The Authors):**
This has the potential to be a valuable tool for the community if it is efficiently distributed. If the authors have not yet done so they should make their plasmids available to the community via Addgene.

We have started the deposit process with Addgene and plasmids will be available soon. In the meantime, all plasmid maps are available on our website https://www.leishbaseedit.net/ and can be requested for shipment from our lab.

Line 162-165, 400-401: The potential for using AsCAS12a's intrinsic RNase activity for "multiplexing" would benefit from a little more explanation (i.e. how this would work, and what multiplexing means in this context).

We have added further details on multiplexing with Cas12a and point out potential applications.

“For example, Cas12a crRNA arrays with four or more guides can be assembled and transfected to introduce multiple DSBs within one gene. Since Cas12a generates sticky DNA ends that facilitate recombination via microhomology-mediated end joining and homologous recombination (Zhang et al., 2021), this approach could effectively disrupt target genes without requiring the addition of donor DNA and this may provide an alternative approach to our here presented base editing method in the future. Moreover, CBE sgRNAs could be multiplexed by interspacing them with Cas12a direct repeats (DRs), enabling simultaneous targeting of multiple genes in one cell.”

Line 193-194: can the authors offer an explanation for the reduction in mNG editing observed with 30nt homology flanks?

We assume this is caused by imprecise recombination events in some cells and have revised the original sentence.

In several places in the manuscript, it is unclear if an analysis has been done on an individual clone or a population derived from multiple transfected cells. If on mixed population, clarify this and calculate the number of clones that the mixture represents. E.g. lines 195-196 and 221-223 (Sanger sequencing of integration site); Line 333-352 (ONT analysis of CBE expression cassette integration).

Only when we tested whether multiple CBE sgRNAs are integrated, we generated and analysed clones (Figure 4 S3). In all other experiments we analysed parasite populations. For better clarity, we have where possible indicated this in the revised manuscript (e.g. at the lines requested).

Line 259: "site by site" should presumably be "side by side".

Many thanks for pointing this out. We have changed this typo.

Lines 315-317: Clarify why the mis-integration of the CBE sgRNA expression cassette might cause a lack of editing (e.g. lack of expression?).

We have added: “This could potentially result in the silencing of the CBE sgRNA expression or even lead to the deletion of the guide cassette”

Line 364 - 367: it is unlikely there is the statistical power to state that 2/10 represents lower than the previously observed 38% of double integrants.

We agree that the statistical power is low and have therefore changed our phrasing to an overall estimation.

**Reviewer #3 (Recommendations For The Authors):**
I suggest that the authors make clearer to the reader the evidence for improved editing efficiency in the new CBE system described here relative to the system described in Engstler and Beneke, 2023. Such clarification could be as simple as an extra paragraph or figure, clearly comparing the editing rates with the two systems in, as far as possible, equivalent conditions.

We have directly compared our improved method to our previous base editing method in Figures 1E and 4, demonstrating higher editing rates in a much shorter time. Especially the L. major panel in Figure 4B shows that in a direct comparison between the previously published (Engstler and Beneke, eLife 2023) and new system, editing can be only observed with the version presented here. However, to clarify the improvements we made, we compare now data from our previous screen done in Engstler and Beneke, eLife 2023 with a loss-of-function screen carried out with our updated method (see Figure 5 and section “Detection of fitness-associated phenotypes in a pooled loss-of-function screen”).

The significance of this work would be improved by running the type of loss of fitness screen described previously in Engstler and Beneke (2023), thereby showing that the new approach improves on previous tools. Without such data, questions remain about potential confounding effects that might not be anticipated from the targeted experiments provided in the current manuscript.

We thank the reviewer for this suggestion. The requested experiment is now presented in Figure 5 and described in section “Detection of fitness-associated phenotypes in a pooled loss-of-function screen”.